# DAL: A Practical Prior-Free Black-Box Framework for Piecewise Stationary Bandits

## Abstract

We introduce a practical, black-box framework termed Detection Augmented Learning (DAL) for the problem of piecewise stationary bandits without knowledge of the underlying non-stationarity. DAL accepts any stationary bandit algorithm with order-optimal regret as input and augments it with a change detector, enabling applicability to all common bandit variants. Extensive experimentation demonstrates that DAL consistently surpasses current state-of-the-art methods across diverse non-stationary scenarios, including synthetic benchmarks and real-world datasets, underscoring its versatility and scalability. We provide theoretical insights into DAL's strong empirical performance, complemented by thorough experimental validation.

## 1 Introduction

Bandit models underpin a wide range of engineering systems, from recommendation and ads to dynamic pricing and real-time bidding (Lefortier et al., 2014; Li et al., 2010; Schwartz et al., 2017; Sertan et al., 2012; Tajik et al., 2024; Flajolet & Jaillet, 2017). Many variants of bandits have emerged since the work of (Robbins, 1952), which fall into parametric bandits (PB) (Auer, 2002; Faury et al., 2020; Filippi et al., 2010), non-parametric bandits (NPB) (Srinivas et al., 2010) and contextual bandits (CB) (Woodroofe, 1979; Langford & Zhang, 2007). In the general bandit problem, in each round, an agent receives a context $C_t$ randomly sampled from a set $\mathcal{C}$, and selects a policy $\pi_t$ from a policy set $\Pi$—a set of mappings from $\mathcal{C}$ to a compact action set $\mathcal{A} \subseteq \mathbb{R}^d$. Then, the agent chooses action $A_t = \pi_t(C_t)$ and receives reward

$$X_t = f_t(C_t, A_t) + \varepsilon_t,$$

where $f_t : \mathcal{C} \times \mathcal{A} \to \mathbb{R}$ is the reward function and $\varepsilon_t$ is the zero-mean sub-Gaussian noise. The goal is to minimize the dynamic regret, using a causal policy $\pi_t$ based on past interactions:

$$R_T := \mathbb{E}_{\substack{A_t \sim \pi_t \\ C_t \sim \mathcal{P}_t}} \left[ \sum_{t=1}^{T} \max_{\pi \in \Pi} f_t(C_t, \pi(C_t)) - f_t(C_t, A_t) \right].$$

CBs follow the general formulation above, where the context $C_t$ is independently sampled from $\mathcal{P}_t$ and $|\mathcal{A}|$ is finite. In PB and NPB settings, the context is fixed across time and $|\mathcal{A}|$ can be infinite. With slight abuse of notation, we write $f_t(C_t, A_t) = f_t(A_t)$ in PBs and NPBs. For PBs, $f_t(A_t) = \mu(\langle \theta_t, A_t \rangle)$, where $\theta_t$ is a bounded unknown parameter and $\mu : \mathbb{R} \to \mathbb{R}$ is injective. These include linear bandits (LBs), with $\mu$ as identity, generalized linear bandits (GLBs), and self-concordant bandits (SCBs), where $\mu$ is self-concordant and the noise variance may depend on the mean (Russac et al., 2021). For NPB, we consider kernelized bandits (KBs), where $f_t \in H_k$, a reproducing kernel Hilbert space (RKHS) induced by a continuous positive semi-definite kernel $k : \mathcal{A} \times \mathcal{A} \to \mathbb{R}$ with $k(x, x) \leq 1$ and $\|f_t\|_{H_k} \leq B$. In KBs, a central complexity measure is the maximum information gain $\gamma_T$ (worst-case mutual information between $f$ and $T$ noisy evaluations). For compact $\mathcal{A} \subset \mathbb{R}^d$: $\gamma_T = \mathcal{O}((\log T)^{d+1})$ for the Squared Exponential (SE) kernel, and $\gamma_T = \mathcal{O}(T^\beta \log T)$ with $\beta = d(d+1)/[2\nu + d(d+1)]$ for Matérn($\nu$) kernels.

Bandits remain practically relevant today: recent deployments span A/B testing (Zhang et al., 2025), clinical trials (Varatharajah & Berry, 2022), large language models (Shin et al., 2025), diffusion models (Aouali, 2024), and computer architecture (Gerogiannis & Torrellas, 2023), which even

leverage the canonical formulations as the core decision engine. Accordingly, the key challenge is developing bandit methods that perform reliably under real-world constraints—aimed at practical effectiveness, not just analysis. The lion's share of the literature on bandits assumes *stationarity*—i.e., fixed $f_t, \theta_t, \mathcal{P}_t$—but this rarely holds in practice due to evolving conditions (Agrawal & Jia, 2019; Cai et al., 2017; Chen et al., 2020; Lu et al., 2019). Non-stationary (NS) settings are often categorized into two types–*gradual drifts* and *abrupt changes*. In the drifting model, $f_t$ and $\mathcal{P}_t$ evolve slowly under a variation budget constraint (Besbes et al., 2014; Wei & Luo, 2021). In contrast, piecewise stationary (PS) models assume abrupt shifts at unknown change-points:

$$1 =: \nu_0 < \nu_1 < \cdots < \nu_{N_T} < \nu_{N_T+1} := T+1, \quad N_T : \text{total number of changes}$$

with $f_t = f_{t'}$ and $\mathcal{P}_t = \mathcal{P}_{t'}$ for $t, t' \in \{\nu_k, \ldots, \nu_{k+1} - 1\}$ and different across change-points.

NS bandit algorithms are typically either *adaptive*—adjusting continuously, or *restarting*—choosing to unlearn and kickstart the learning process at certain times. They may also be *prior-based* (assuming knowledge of the non-stationarity) or *prior-free*. Prior-based adaptive methods (discounting/sliding window) weigh recent observations more heavily: NS multi-armed bandits (NS-MABs) (Garivier & Moulines, 2011; Kocsis & Szepesvári, 2006), NS-LBs (Cheung et al., 2019; Russac et al., 2019), NS-GLBs (Faury et al., 2021; Russac et al., 2020), NS-SCBs (Russac et al., 2021; Wang et al., 2023), NS-KBs (Deng et al., 2022; Zhou & Shroff, 2021). Prior-based restarting approaches use budgeted restarts: NS-MABs (Besbes et al., 2014), NS-LBs/GLBs (Zhao et al., 2020), NS-KBs (Zhou & Shroff, 2021). Detection-based restarting methods exist in both flavors: prior-based for NS-MABs (Cao et al., 2019b; Liu et al., 2018) and NS-CBs (Luo et al., 2018); prior-free for NS-MABs (Auer et al., 2019; Besson et al., 2022; Huang et al., 2025), for NS-LBs/KBs (Hong et al., 2023) and for NS-CBs (Chen et al., 2019). The most closely related work is Huang et al. (2025), which addresses PS-MABs and introduces techniques that we build upon in establishing our theory.

Among prior-free methods, *black-box* approaches are particularly appealing: they equip *any* stationary bandit algorithm with non-stationarity handling capabilities. MASTER (Wei & Luo, 2021) is the only known order-optimal black-box method for general bandit and reinforcement learning settings. Importantly, although MASTER is order-optimal, it is not practically applicable (Gerogiannis et al., 2025). More broadly, the literature emphasizes theory over evidence, as empirical validation of order-optimal methods is scarce: NS-NPBs and NS-PBs are evaluated almost exclusively on synthetic data (Wang et al., 2023; Hong et al., 2023; Gerogiannis et al., 2025), and NS-CBs lack experiments altogether (Chen et al., 2019). We close these gaps with a theoretically grounded, practical black-box framework and comprehensive real-world evaluation in standard benchmarks.

**Contributions.** We present (to our knowledge) the first *practical* prior-free, black-box detection-based framework for general PS bandits. The design is motivated by three pragmatic insights: (i) prior knowledge of non-stationarity is rarely available, (ii) restart-style methods can have lower worst-case complexity than fully adaptive schemes (Peng & Papadimitriou, 2024), and (iii) a black-box reduction simplifies NS algorithm design to specifying when to restart a stationary learner. Our method is simple—combining a change detector with any stationary bandit algorithm—modular, and easy to implement. Empirically, extensive synthetic and real-world evaluations in standard datasets show consistent gains over both prior-free and prior-based baselines, and (to our knowledge) provide the first comprehensive real-world assessment of order-optimal baselines previously lacking empirical study. Theoretically, under mild assumptions, our regret matches the state-of-the art for PS-LBs, PS-GLBs and PS-CBs and *improves* the best known bounds for PS-SCBs and PS-KBs; for drifting regimes we identify conditions for good performance and validate them empirically.

## 2 THE DAL FRAMEWORK

The DAL framework is a black-box characterized by a modular structure of three components: a non-stationarity detector, a forced exploration scheme, and a bandit algorithm. We provide high-level ideas of the structure of our approach and formally present our framework in Algorithm 1.

**Non-Stationarity Detector** To identify changes in the environment, DAL uses a general-purpose detector $\mathcal{D}$ for monitoring shifts in the distribution of judiciously chosen reward observation sequences obtained through forced exploration. This distinguishes our approach from methods like

MASTER, which rely on detecting violations of stationary regret guarantees. We adopt a detector aligned with Besson et al. (2022); Huang et al. (2025), grounded in the well-established theory of quickest change detection (Veeravalli & Banerjee, 2013; Xie et al., 2021). Given any arbitrary context, DAL samples rewards from actions within a carefully selected finite subset, and detects changes in the mean reward associated with the context-action pair.

---

**Alg. 1** **D**etection **A**ugmented **L**earning (**DAL**)

---

**Input**: bandit $\mathcal{B}$, detector $\mathcal{D}$, covering set size $N_e$, covering set $\mathcal{A}_e = \{a^{(i)} : i \in [N_e]\}$, context set $\mathcal{C}$, frequencies $\{\alpha_k\}_{k=1}^T$, horizon $T$
**Initialize**: histories $\mathcal{H}_{(c,a)} \leftarrow \emptyset \ \forall (c,a) \in \mathcal{C} \times \mathcal{A}_e$, detection $\tau \leftarrow 0$, counter $k \leftarrow 1$

1: **for** $t = 1, 2, \ldots, T$ **do**
2:    Observe context $C_t$
3:    **if** $(t - \tau + 1 \mod \lceil N_e/\alpha_k \rceil) + 1 = i \in [N_e]$ **then**
4:       Play action $a^{(i)}$ and receive reward $X_t$
5:       Add reward $X_t$ into history $\mathcal{H}_{(C_t, a^{(i)})}$
6:       **if** $\mathcal{D}\left(\mathcal{H}_{(C_t, a^{(i)})}\right) = $ detection **then**
7:          Reset the bandit algorithm $\mathcal{B}$
8:          Clear all $\mathcal{H}_{(c,a)} \ \forall (c,a) \in \mathcal{C} \times \mathcal{A}_e$,
9:          $\tau \leftarrow t, \quad k \leftarrow k + 1$
10:      **end if**
11:    **else**
12:       Run the stationary bandit algorithm $\mathcal{B}$
13:    **end if**
14: **end for**

---

**Forced Exploration** In stationary bandit settings, effective algorithms quickly concentrate on (near-)optimal actions for each context, rarely exploring suboptimal actions. In NS environments, however, this behavior may lead to missed changes on these rarely sampled actions, and thus, *forced exploration* on these actions is essential. When the action space is large or infinite, exploring all actions becomes infeasible. Therefore, DAL only does extra exploration on a finite *covering set*, $\mathcal{A}_e = \{a^{(i)} : i \in [N_e]\} \subseteq \mathcal{A}$, in which $a^{(i)}$ denotes the $i$-th action in $\mathcal{A}_e$. $\mathcal{A}_e$ is designed such that the mean reward of at least one context-action pair in $\mathcal{C} \times \mathcal{A}_e$ changes whenever a change occurs. In particular, after the $(k-1)^{\text{th}}$ restart, DAL is forced to play each action in $\mathcal{A}_e$ once for $N_e$ steps, followed by the bandit algorithm for the next $\lceil N_e/\alpha_k \rceil - N_e$ steps, repeatedly, until the $k^{\text{th}}$ restart. Here, $\alpha_k \in (0, 1)$ is the exploration frequency, striking a balance between detection delay and regret from extra exploration.

**Bandit Algorithm** With a detector $\mathcal{D}$ and forced exploration, DAL augments a (stationary) bandit algorithm $\mathcal{B}$: It resets $\mathcal{B}$ entirely whenever $\mathcal{D}$ detects changes in a reward distribution associated with any context-action pair in $\mathcal{C} \times \mathcal{A}_e$ (Line 6), and runs $\mathcal{B}$ with periodic forced exploration otherwise. Essentially, the purpose of the detector is to identify shifts in the mean of the rewards, i.e., changes in the environment. Line 6 and $\mathcal{D}$ are fully elaborated in Sections 3, 4.1 and in the Appendix. A key advantage of DAL is its ability to translate strong stationary performance into robust performance under NS conditions. Therefore, by selecting a well-performing bandit algorithm, the DAL framework inherently achieves effective adaptation to NS environments. In fact, the only requirement for DAL's input stationary algorithm is to attain optimal stationary regret performance bounds.

## 3 PRACTICAL PERFORMANCE

### 3.1 EXPERIMENTAL BASELINES

In our experiments, we evaluate *all* methods referenced in Section 1, highlighting the strongest state-of-the-art algorithms applicable to PS and drifting settings across both synthetic and real-world benchmarks. These baselines include MASTER (Wei & Luo, 2021), the only other black-box method with order-optimal regret. MASTER lacks guarantees for SCBs, but empirical evidence (Wang et al., 2023) supports using it with Log-UCB-1 (Faury et al., 2020) in NS-SCBs. We additionally include two prior-free, order-optimal algorithms: ADA-OPKB (Hong et al., 2023) for NS-LBs/NS-KBs and ADA-ILCTB+ (Chen et al., 2019) for NS-CBs. ADA-OPKB requires extensive tuning (7 hyperparameters), which is incompatible with a fully prior-free setting; nevertheless, we tune it (and MASTER's single parameter $n$) for best performance in our evaluation. We also compare against two prior-based discounted approaches—WeightUCB (Wang et al., 2023) for drifting PBs and PS-SCBs, and WGP-UCB (Deng et al., 2022) for drifting KBs. All remaining methods are *prior-based*. To maintain figure readability, we group algorithms by paradigm (discounted, sliding-window, budget-restart) while keeping distinct methods separate when they differ meaningfully. In real-world experiments, we focus on the strongest current state-of-the-art methods, as the remaining algorithms are less competitive. We use the hyperparameters specified in the original works.

## 3.2 PRACTICAL TUNING OF DAL

Across all settings, DAL uses the *Generalized Likelihood Ratio (GLR)* and the *Generalized Shiryaev-Roberts (GSR)* tests (Huang & Veeravalli, 2025) as the detector $\mathcal{D}$, which are given in Algorithms 2 and 3. For the detectors, we set their thresholds $\beta_{\text{GLR}}(n, \delta_F) = \log(n^{3/2}/\delta_F)$ and $\beta_{\text{GSR}}(n, \delta_F) = n^{5/2}/\delta_F$, with $\delta_F = 1/\sqrt{T}$, as per Huang et al. (2025); Besson et al. (2022). Concretely: In NS-LBs, LinUCB (Abbasi-yadkori et al., 2011) pairs with Gaussian GLR and GSR. In NS-GLBs, GLM-UCB (Filippi et al., 2010) pairs with Gaussian GLR and GSR. In NS-SCBs, OFUGLB (Lee et al., 2024) pairs with Bernoulli GLR and GSR. In NS-KBs, REDS (Salgia et al., 2024) pairs with Gaussian GLR and GSR. In NS-CBs, SquareCB (Foster & Rakhlin, 2020) pairs with Bernoulli GLR and GSR. We implement the stationary bandit algorithms as provided in their original works. For all settings, we set $\alpha_k = \sqrt{k||\mathcal{C}|N_{\text{e}}}/(2\sqrt{T}\log^2 T)$ as per Theorem 4.8. A crucial advantage of DAL is that it is *hyperparameter-free*, guided entirely by our theoretical principles.

To construct $\mathcal{A}_e$, for NS-PBs we follow Proposition 4.2: we greedily select linearly independent actions until collecting $d$ such vectors, or as many as exist if fewer than $d$ are available. In our experiments, actions are sampled from a multivariate Gaussian, which always yields $d$ such vectors. For NS-KBs, $\mathcal{A}_e$ is from a $\delta_T$-cover by selecting the centers of the covering balls according to Proposition 4.3 and Corollary 4.9. In finite action spaces, we compute $\gamma_T$; if $|\mathcal{A}| \leq \gamma_T$, then by Corollary 4.9 the action set already forms a valid cover and we take $\mathcal{A}_e = \mathcal{A}$, otherwise we select the $\gamma_T$ actions closest to the cover centers. In all our NS-KB experiments, $\gamma_T$ is larger than $|\mathcal{A}|$, so we always have $\mathcal{A}_e = \mathcal{A}$. In PS-CBs, the action space is finite, and as noted in Remark 4.4, both the theory and our experiments take $\mathcal{A}_e = \mathcal{A}$. DAL's exploration burden is $N_e = |\mathcal{A}_e|$, which is determined by the structural complexity of the reward class. In PBs and KBs $|\mathcal{A}_e|$ is independent of the infinite $\mathcal{A}$, while in CBs all *finite* actions must be explored. DAL limits $N_e$ to the minimum needed to characterize the reward function for detection and learning. DAL is driven by structural complexity, not $|\mathcal{A}|$. An extended discussion on $\mathcal{A}_e$ and its implications appears in the Appendix.

## 3.3 SYNTHETIC EXPERIMENTS

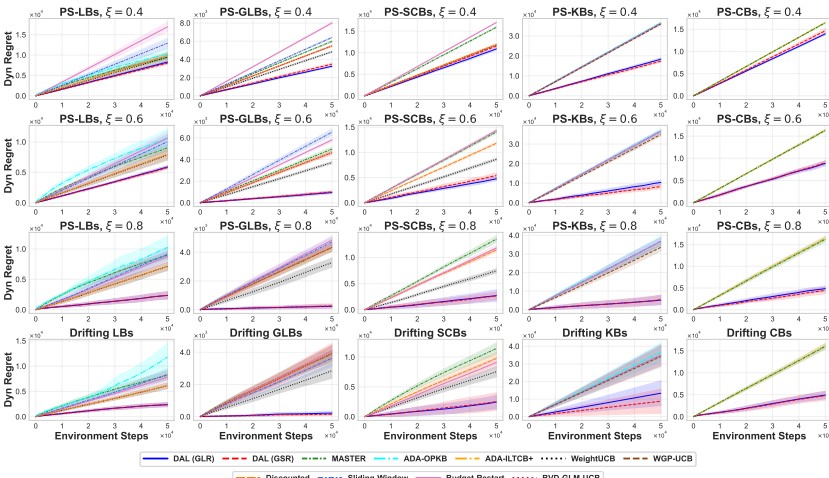

Figure 1: Dynamic regret vs. environment steps for synthetic experiments (lower=better). First three rows correspond to the geometric change-points and the final one to the drifting case.

### 3.3.1 EXPERIMENTAL PARAMETERS

**Common parameters** In all synthetic experiments, the action space comprises 100 unique actions with dimension $d = 10$. These actions are sampled independently from $\mathcal{N}(0, I)$. The horizon is fixed to $T = 50000$ and we average the results over 15 independent trials.

**Remark 3.1.** *In Algorithm 1, when $|\mathcal{A}|$ is finite, change-detection can be performed on the actions selected by $\mathcal{B}$ that are not in $\mathcal{A}_e$, which improves performance. This does not affect the theoretical properties of the algorithm, and we employ this variation for our experiments.*

**NS-PBs** The actions are scaled to lie within an $L$-ball and the underlying parameters $\theta_t$ belong to an $S$-ball. Specifically, for NS-LBs and NS-GLBs we have that $S = L = 1$, while for NS-SCBs, we have that $L = 1$ but $S = 3$. Every time a $\theta_t$ is initialized or changed, its elements are chosen independently and uniformly in $[-1, 1]$, and then are scaled to the $S$-ball. For both NS-GLBs and NS-SCBs, we select $\mu(x) := \sigma(x) = (1 + e^{-x})^{-1}$ (sigmoid). The additive noise $\varepsilon_t$ is sampled according to $\mathcal{N}(0, 0.01)$ at each time-step, while for NS-SCBs, we sample the random reward according to $\text{Bernoulli}(\mu(\langle \theta_t, A_t \rangle))$ at time $t$. To set $\mathcal{A}_e$ in NS-PBs, we use Corollary 4.9.

**NS-KBs** Actions are scaled in the $\sqrt{d}$-ball and $\varepsilon_t \sim \mathcal{N}(0, 0.01)$. We employ the SE kernel with $\ell = 0.2$. We follow a procedure similar to Chowdhury & Gopalan (2017); Deng et al. (2022). Specifically, every time we initialize or change the reward function, $f_t$ is generated from the RKHS obtained by a discretization of $[-1, 1]$ into 200 evenly spaced points. The reward function is set as $f(\cdot) = \sum_{i=1}^{M} \alpha_i k(\cdot, x_i)$, $\alpha_i \sim \text{Unif}[-1, 1]$ and $M = 200$. For $\mathcal{A}_e$, we use Corollary 4.9.

**NS-CBs** The context $C_t \in \mathbb{R}^{d_c}$ is drawn at each round from a fixed pool of 1000 normalized vectors with $d_c = 10$, according to a categorical distribution. At every initialization or change, at least one of the $f_t$ or $\mathcal{P}_t$ changes, while $\Pi$ is fixed in each run. For $a \in \mathcal{A}$ and context $C_t$, $f_t$ is clipped in $[0, 1]$, and is given by

$$f_t(C_t, a) = \left[ b_a + z^{(\text{sig})} \sigma(u_a^\top C_t) + z^{(\text{sin})} \sin(v_a^\top C_t) + z^{(\text{xpr})} C_{t,2} C_{t,3} \right]_{[0,1]},$$

where $u_a, v_a \sim \mathcal{N}(0, I)$, $b_a \sim \text{Unif}[0.3, 0.7]$, and $z^{(\text{sig})}, z^{(\text{sin})}, z^{(\text{xpr})}$ are drawn uniformly from $[0.25, 0.45]$, $[0.15, 0.35]$, $[0.10, 0.25]$, respectively. Rewards are sampled as $\text{Bernoulli}(f_t(C_t, A_t))$. Since the reward function lacks any arm-related structure, here we set $\mathcal{A} = \mathcal{A}_e$ (see Remark 4.4).

### 3.3.2 EXPERIMENTAL BENCHMARKS

**Piecewise Stationarity** In the PS setting, we adopt the geometric change-point model proposed in Gerogiannis et al. (2025), and independently sample the intervals between the change-points according to a geometric distribution with parameter $\rho = T^{-\xi}$, for $\xi \in \{0.4, 0.6, 0.8\}$. We do not impose any restriction on the lengths of the intervals between change-points in our experiments.

**Drifting Non-Stationarity** Regarding comparisons in drifting non-stationarity, we adopt the following drift model: in each run, the reward structure changes linearly over $T$ rounds from an initial value to a final value, where the end-points are chosen as in the beginning of the section. Specifically,

$$\text{PBs: } \theta_t = (1 - t/T) \, \theta_{\text{init}} + (t/T) \, \theta_{\text{final}}, \quad \text{KBs: } f_t = (1 - t/T) \, f_{\text{init}} + (t/T) \, f_{\text{final}},$$

$$\text{CBs: } \phi_t = (1 - t/T) \, \phi_{\text{init}} + (t/T) \, \phi_{\text{final}}, \; \phi_t := (u_{a,t}, v_{a,t}, b_{a,t}, \mathbf{z}_t), \; \mathbf{z}_t := (z_t^{(\text{sig})}, z_t^{(\text{sin})}, z_t^{(\text{xpr})}).$$

**Experimental Results** Per the results in Figure 1, DAL outperforms the current state-of-art methods in every synthetic experiment for both choices of detectors. DAL only abandons the actions chosen by the stationary bandit algorithm and restarts learning when an efficient change detector flags a mean-shift in rewards; hence, it avoids unnecessary restarts, especially when the intervals between the change-points are long enough for such detectors to correctly flag said changes without false alarms. Regarding drifting non-stationarity, DAL significantly outperforms all other methods. In fact, it fares better than both WeightUCB and ADA-OPKB, which not only are known to attain the optimal regret bound in the drift setup, but have also been shown to perform well in practice.

### 3.4 REAL-WORLD EXPERIMENTS

**Microarchitecture Prefetcher Selection Benchmark.** We introduce a novel dataset for NS bandit evaluation using the data of Gerogiannis & Torrellas (2023).[1] The dataset includes 11 prefetcher configurations (actions) that trade aggressiveness against efficiency. At each time-step, the reward is the normalized instructions per cycle in $[0, 1]$, and the horizon is $T = 26224$. Following Gerogiannis & Torrellas (2023), we also evaluate D-UCB (Kocsis & Szepesvári, 2006) in its native form; while for our baselines we model the task as an NS-SCB. D-UCB hyperparameters follow its original paper and Gerogiannis & Torrellas (2023). Evaluation is by cumulative reward.

---

[1]We aim to release the dataset to facilitate real-world experimentation by the bandit research community.

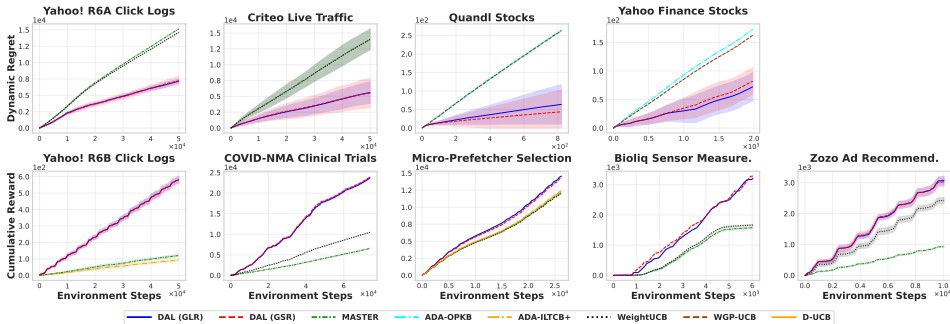

Figure 2: Results for real-world experiments of Section 3.4, averaged over 15 independent runs. Top: dynamic regret (lower=better); Bottom: cumulative reward (higher=better).

**Stock Market Benchmarks.** NS-KBs have been applied to stock market prediction, and we follow the procedure of (Deng et al., 2022) to simulate two environments: one using their original data (Quandl stocks) and one constructed from NASDAQ-100 stocks retrieved via the `yfinance` Python package.[2] In the Yahoo-based dataset, we retain stocks with sufficient history ($T$=2000, approx. 5.5 years) and select the 50 most volatile as actions. Daily closing prices define the reward function, and the empirical price covariance matrix is used as the kernel. To increase difficulty, we add Gaussian noise $\mathcal{N}(0, 0.01)$ to the reward at each time-step. Evaluation is by dynamic regret.

**COVID-NMA Clinical Benchmark.** We construct an NS-SCB benchmark from the open COVID-NMA database (Boutron et al., 2025). To maximize coverage while retaining clinical meaning, we form a *UNION* endpoint: for each bucketed-treatment arm and month, we include both Clinical Improvement at Day 28 (when reported) and Survival at Day 28 (1–mortality) as separate contributions, leading to binary rewards (1=success). Treatments (actions) are mapped into 13 classes and month counts are expanded exactly ($s$ successes and $n-s$ failures per bin) and concatenated in a fixed chronological order (month, `clinD28` then `mortD28`, then bucket) to yield a long non-stationary sequence with $T \approx 7.4 \times 10^4$. Evaluation is based on cumulative reward.

**Click Log Benchmarks.** We use the Yahoo! R6A click log dataset.[3] Following (Cao et al., 2019b; Seznec et al., 2020), we compute smoothed click-through rates (CTRs) via rolling averages over 2000 rounds, average CTRs within each subperiod, and suppress fluctuations below 0.005. We combine actions across 5 days, leading to 64 actions, compress the horizon to 50000, and multiply final CTRs by 10. We model the resulting environment as an NS-SCB problem, reflecting the logistic reward structure typical in such settings (Russac et al., 2021). Evaluation is by dynamic regret.

Alongside the R6A, we build a fixed-arm replay benchmark from the additional Yahoo! R6B click logs, as an NS-CB problem.[3] We select the highest-CTR articles to form an action set of 51 actions, round-robin interleave days and then select $T = 50000$. For each visit, we intersect the candidate set with this vocabulary, keep rounds where the displayed item remains, and record the binary click as the raw reward ($X_t \in \{0, 1\}$). We rely on R6B's uniform-random logging for unbiased replay/IPS evaluation (Li et al., 2011). Our metric is (replay) cumulative reward.

**Live Traffic Benchmark.** We construct an NS bandit environment based on the Criteo live traffic dataset (Diemert et al., 2017), following the preprocessing approach of Russac et al. (2019) but modeling the problem as an NS-GLB rather than an NS-LB. We estimate the underlying parameter $\theta^*$ using logistic regression. Unlike Russac et al. (2019), in which the authors employ a single change, we introduce shifts in $\theta^*$ via a geometric change-point model with parameter $\xi = 0.8$ and extend the horizon to $T = 50000$. The metric here is the dynamic regret.

**Sensor Correlation Benchmark.** We use the Bioliq dataset provided by Komiyama et al. (2024), which contains a week of measurements from 20 sensors in a power plant. We process the reward as Komiyama et al. (2024) and construct an NS-SCB environment with 190 actions. At each time-

---

[2]Data retrieved from Yahoo Finance using the publicly available `yfinance` package. Used solely for non-commercial, academic research purposes.

[3]Yahoo! Front Page Today Module User Click Log Datasets: https://webscope.sandbox.yahoo.com.

step, the reward is 1 if the last 1000 measurements exceed a threshold of 2.04, and 0 otherwise. Evaluation is based on cumulative reward.

**Ad Recommendation Benchmark.** We evaluate on the Zozo environment, a real-world ad recommender system deployed on an e-commerce platform, introduced by Saito et al. (2021). Using the dataset preprocessed by Komiyama et al. (2024), we construct an NS-GLB environment that captures the dynamics of online ad recommendation. Unlike Komiyama et al. (2024), in which the authors limit the setup to 10 actions due to sparsity, we keep all 80 ads as actions. Following their setup, we assign a reward of 1 to any ad that received at least one user click within a one-second window, and 0 to ads with no clicks. Here, evaluation is based on cumulative reward.

Based on the results in Figure 2, DAL consistently outperforms all state-of-the-art baselines across real-world benchmarks, in both dynamic regret and cumulative reward with both GLR and GSR. We attribute this strong performance to the robustness DAL demonstrates in the synthetic settings, which captured a range of challenging NS scenarios. These findings underscore DAL's practical effectiveness. In what follows, we provide a theoretical explanation for its performance.

## 4 THEORETICAL INSIGHTS

### 4.1 ON EFFECTIVE DETECTION

When selecting a non-stationarity detector, accuracy and efficiency are essential for ensuring optimal regret growth. Any detector aiming to identify distribution shifts inherently requires a certain number of samples, both before and after the change. Ideally, this sample complexity should scale appropriately to avoid negatively impacting the total regret. To this end, the GLR and GSR tests have been shown to achieve a pre- and post-change sample complexity of the order $\log T$ (Huang & Veeravalli, 2025). Since logarithmic terms are disregarded in dynamic regret analyses, it suggests that integrating such detection mechanisms could achieve optimal regret growth.

The stopping time $\tau$ of a change detector $\mathcal{D}$ denotes the time-step at which a change is identified. Let $\mathbb{P}_\nu$ and $\mathbb{E}_\nu$ be the probability and expectation with change-point at $\nu$, and $\mathbb{P}_\infty$ and $\mathbb{E}_\infty$ be the ones with no change-point. The *latency* $\ell_\mathcal{D}$ is the length of time post-change within which a change is declared with a probability $1 - \delta_{\mathrm{D}}$, i.e.,

$$\ell_\mathcal{D} := \inf\{t \in [T] : \mathbb{P}_\nu(\tau \geq \nu + t) \leq \delta_{\mathrm{D}}, \ \forall \nu \in [m_\mathcal{D} + 1, T - t]\}$$

where $m_\mathcal{D}$ is the length of the pre-change window at which no changes occur. A good detector seeks to minimize $\ell_\mathcal{D}$ while ensuring low false-alarm probability over horizon $T$, namely $\mathbb{P}_\infty(\tau \leq T) \leq \delta_{\mathrm{F}}$ with $\delta_{\mathrm{F}} \in (0, 1)$. To ensure order-optimal regret for DAL, the detector $\mathcal{D}$ must satisfy:

**Property 4.1.** $\ell_\mathcal{D} + m_\mathcal{D} = \mathcal{O}(\log T + \log(1/\delta_{\mathrm{F}}) + \log(1/\delta_{\mathrm{D}}))$.

This condition is crucial in the proof of Theorem 4.8 in the Appendix. We employ the GLR and GSR tests since they satisfy Property 4.1 (Huang & Veeravalli, 2025), with thresholds $\beta_{\mathrm{GLR}}(n, \delta_{\mathrm{F}}) = \mathcal{O}(\log(n^{3/2}/\delta_{\mathrm{F}}))$ and $\beta_{\mathrm{GSR}}(n, \delta_{\mathrm{F}}) = \mathcal{O}(n^{5/2}/\delta_{\mathrm{F}})$. In experiments, GSR performs slightly better than GLR, but the difference is minor since both satisfy Property 4.1. This shows that the good performance is due to the *design* of DAL rather than the specifics of a single detector.

| **Alg. 2 Generalized Likelihood Ratio Test** | **Alg. 3 Generalized Shiryaev–Roberts Test** |
|---|---|
| **Input**: History $\mathcal{H} = \{X_1, \ldots, X_n\}$, $\delta_{\mathrm{F}}$, $\delta_{\mathrm{D}}$, KL divergence $\mathrm{kl}(\cdot, \cdot)$ | **Input**: History $\mathcal{H} = \{X_1, \ldots, X_n\}$, $\delta_{\mathrm{F}}$, $\delta_{\mathrm{D}}$, KL divergence $\mathrm{kl}(\cdot, \cdot)$, $\mathrm{GSR}_k \leftarrow 0$ |
| 1: **for** $k = 1$ to $n - 1$ **do** | 1: **for** $k = 1$ to $n - 1$ **do** |
| 2:    Compute empirical means $\hat{\mu}_{1:k}$, $\hat{\mu}_{k+1:n}$, $\hat{\mu}_{1:n}$ | 2:    Compute empirical means $\hat{\mu}_{1:k}$, $\hat{\mu}_{k+1:n}$, $\hat{\mu}_{1:n}$. |
| 3:    $\mathrm{GLR}_k \leftarrow k \, \mathrm{kl}(\hat{\mu}_{1:k}, \hat{\mu}_{1:n})$ | 3:    Compute $\mathrm{GLR}_k$ according to Alg. 2 |
| 4:          $+ (n - k) \, \mathrm{kl}(\hat{\mu}_{k+1:n}, \hat{\mu}_{1:n})$ | 4:     $\mathrm{GSR}_k \leftarrow \mathrm{GSR}_k + \exp(\mathrm{GLR}_k)$ |
| 5:    **if** $\mathrm{GLR}_k \geq \beta_{\mathrm{GLR}}(n, \delta_{\mathrm{F}})$ **then** | 5:    **if** $\mathrm{GSR}_k \geq \beta_{\mathrm{GSR}}(n, \delta_{\mathrm{F}})$ **then** |
| 6:      **return** detection | 6:      **return** detection |
| 7:    **end if** | 7:    **end if** |
| 8: **end for** | 8: **end for** |

The Bernoulli GLR and GSR are used for sub-Bernoulli rewards with $\mathrm{kl}(x, y) = x \ln(x/y) + (1 - x) \ln\left(\frac{1-x}{1-y}\right)$, and the Gaussian variants are for $\sigma^2$-sub-Gaussian rewards with $\mathrm{kl}(x, y) = \frac{(x-y)^2}{2\sigma^2}$.

To select which samples should be fed into the detector, one needs to properly select the covering set $\mathcal{A}_e$, so that it contains actions that can capture changes in the reward function for any context. However, changes cannot be arbitrarily small, as no change detector may be able to identify them. Hence, $\mathcal{A}_e$ should be designed such that whenever a change occurs, reward sequences associated with at least one context-action pair in $\mathcal{C} \times \mathcal{A}_e$ exhibit an *appreciable* mean-shift. Define

$$\Delta_c := \inf_{f \neq f'} \max_{(c,a) \in \mathcal{C} \times \mathcal{A}_e} |f(c,a) - f'(c,a)|.$$

Then, $\Delta_c$ captures how well the context-action pairs in $\mathcal{C} \times \mathcal{A}_e$ can discern between candidate reward functions. According to Huang et al. (2025), $\Delta_c$ crucially affects the performance of the GLR and GSR tests, as their pre- and post- change sample complexity grows with $1/\Delta_c^2$. The more discernible the changes are, the easier the detection becomes. Since forced exploration incurs regret, $\mathcal{A}_e$ should be chosen to minimize $N_e$ while maximizing $\Delta_c$. However, this cannot be done since the function $f_t$ is unknown. Hence, we provide the ways with which one can ensure appreciable mean-shift (i.e., $\Delta_c > 0$) in settings where the reward function has a certain *structure* (e.g., linear dependence on the arms or prescribed smoothness). Specifically, the NS-PB and NS-KB settings satisfy such conditions. Using these choices of $\mathcal{A}_e$, one can guarantee order-optimal regret in certain cases, as shown in the next section. The proofs of the following propositions are given in the Appendix.

**Proposition 4.2.** *In NS-PBs, $\mathcal{A}_e$ can be any arbitrary maximal linearly independent subset of $\mathcal{A}$.*

**Proposition 4.3.** *In NS-KBs, assume that $\mathcal{A} \subseteq [0, R]^d$ w.l.o.g., and that there exists an $\tilde{a} \in \mathcal{A}$ s.t.*

$$\inf_{f \neq f'} |f(\tilde{a}) - f'(\tilde{a})| > L_T,$$

*for some $L_T > 0$. Let $\delta_T := L_T/(2BL_u)$, where $BL_u$ is the Lipschitz constant of all $f \in H_k(\mathcal{A})$ and let $\mathcal{V}_T \subset \mathcal{A}$ be the set of the centers of the balls of an arbitrary $\delta_T$-cover. Then, $\mathcal{A}_e$ can be taken as $\mathcal{V}_T$, with $|\mathcal{V}_T| \leq \lceil \sqrt{d}R/2\delta_T \rceil^d = \lceil \sqrt{d}BL_uR/L_T \rceil^d$.*

**Remark 4.4.** *In NS-CBs, if $f_t$ and $\mathcal{A}$ satisfy the structural assumptions of the preceding propositions for any fixed context, we can set $\mathcal{A}_e$ similarly. Without such structure, we set $\mathcal{A}_e = \mathcal{A}$, as $\mathcal{A}$ is finite.*

## 4.2 ON ORDER-OPTIMALITY IN PIECEWISE STATIONARY ENVIRONMENTS

In the PS setting, the minimax regret lower bound under bandit feedback is $\tilde{\Omega}(\sqrt{N_T T})$ (Garivier & Moulines, 2011),[4] which applies across all settings considered in this work, differing only in problem-dependent constants. Under certain conditions on the minimum spacing between change-points, our algorithm matches this bound with state-of-the-art dependence on these constants. Specifically, the assumption states that $\nu_k - \nu_{k-1}$ should be large enough to acquire enough samples to trigger restarts. For brevity, we first define the relevant quantities and then state the assumption.

**Definition 4.5.** For PS-PBs and PS-KBs, let $m_k := \lceil N_e/\alpha_k \rceil m_{\mathcal{D}}$ and $\ell_k := \lceil N_e/\alpha_k \rceil \ell_{\mathcal{D}}$ for $k \in [N_T]$. For PS-CBs, let $m_k := \lceil N_e/\alpha_k \rceil \lceil m_{\mathcal{D}}/s + \log T/4s^2 + \sqrt{m_D \log(T)/2s^3 + (\log T)^2/16s^4} \rceil$ and $\ell_k := \lceil N_e/\alpha_k \rceil \lceil \ell_{\mathcal{D}}/s + \log(T)/4s^2 + \sqrt{\ell_D \log T/2s^3 + (\log T)^2/16s^4} \rceil$ for $k \in [N_T]$, with $s := \min_{c \in \mathcal{C}, t \in [T]: \mathcal{P}_t(c) > 0} \mathcal{P}_t(c)$.

**Assumption 4.6.** Assume $\nu_1 \geq m_1$ and $\nu_k - \nu_{k-1} \geq \ell_{k-1} + m_k$ for $k \in \{2, \ldots, N_T\}$.

In PS-PBs and PS-KBs, DAL performs round-robin forced exploration on each arm every $\lceil N_e/\alpha_k \rceil$ rounds. Thus, the scaling of $m_{\mathcal{D}}$ and $\ell_{\mathcal{D}}$ in Definition 4.5 is necessary for Assumption 4.6 to guarantee that the change detector in each arm observes at least $m_{\mathcal{D}}$ pre-change samples and $\ell_{\mathcal{D}}$ post-change samples. In PS-CBs, each context–action pair is only seen in expectation (not deterministically) at least once every $\lceil N_e/\alpha_k \rceil/s$ rounds due to randomness. Thus, in Assumption 4.6, we increase the change-point separation, as shown in Definition 4.5, to collect the $m_{\mathcal{D}}$ and $\ell_{\mathcal{D}}$ samples with high probability. These conditions allow $\mathcal{D}$ to reliably detect a change (Property 4.1).

The assumption on the minimum separation between change-points essentially requires scaling as $\tilde{\mathcal{O}}(\sqrt{T/k})$. However, this condition primarily emerges from a conservative proof technique, where missed detections are aggregated into a single adverse event. Practically, and as corroborated by our experiments, this assumption is often violated without negatively impacting the regret performance—even under scenarios with frequent and arbitrarily placed change-points (e.g. $\xi = 0.4$).

---

[4]We use the $\sim$ in $\tilde{\Omega}(\cdot)$ to hide polylogarithmic factors.

We suspect that this resilience arises because any detector satisfying Property 4.1, while potentially missing isolated short intervals, reliably detects subsequent changes when stationary segments exceed the threshold length. Even if a change is entirely missed during a segment shorter than $\tilde{\mathcal{O}}(\sqrt{T/k})$, the resulting regret remains under that order. Conversely, when the assumption holds, optimal regret is provably guaranteed. Thus, the required separation threshold acts as a practical "sweet spot": segments longer than this threshold are detected reliably, ensuring optimal performance, while shorter segments incur minimal regret, thereby preserving overall optimal regret.

**Remark 4.7.** *Assumption 4.6 is necessary to prove the order-optimality, **but it is not for practical performance**. None of our experiments enforced this assumption, and DAL dominated in both the synthetic and the real-world simulations as shown in Section 3.*

Based on Algorithm 1, DAL can incorporate any stationary bandit algorithm. Since different algorithms yield different regret guarantees, DAL attains order-optimal regret in PS environments only when the stationary component has optimal minimax regret, namely $\tilde{\mathcal{O}}(d\sqrt{T})$ in PBs, $\tilde{\mathcal{O}}(\sqrt{\gamma_T T})$ in KBs, and $\tilde{\mathcal{O}}(\sqrt{|\mathcal{A}| \log |\Pi| \, T})$ in CBs. This requirement is formalized in Theorem 4.8.

Thus, to characterize DAL's performance under piecewise stationarity, we employ the methodology of Huang et al. (2025), incorporating the regret analysis of the stationary bandit algorithm and that of the change detector. Since we are studying general bandits, additional novel analysis is required. Due to space constraints, the full analysis and proof of Theorem 4.8 is deffered to the Appendix.

**Theorem 4.8.** *For the PS setting, consider DAL with a detector $\mathcal{D}$ that satisfies Property 4.1, a stationary bandit algorithm $\mathcal{B}$ with regret upper bound $R_\mathcal{B}$ concave and increasing with $T$, a covering set $\mathcal{A}_\mathrm{e}$ and forced exploration frequencies $(\alpha_k)_{k=1}^T$. If Assumption 4.6 holds, $\alpha_k = \sqrt{k|\mathcal{C}|N_\mathrm{e}}/(2\sqrt{T}\log^2 T)$, $\delta_\mathrm{F} = \delta_\mathrm{D} = T^{-\gamma}$, with $\gamma > 1$, and $R_\mathcal{B}(T) = \tilde{\mathcal{O}}(d^p \gamma_T^q (|\mathcal{A}| \log |\Pi|)^r \sqrt{T})$ with $p, q, r \geq 0$, then DAL's regret satisfies, $R_T = \tilde{\mathcal{O}}(d^p \gamma_T^q (|\mathcal{A}| \log |\Pi|)^r \sqrt{N_T T} + \sqrt{|\mathcal{C}|N_\mathrm{e} N_T T})$.*

Using Theorem 4.8, Propositions 4.2, 4.3 and Remark 4.4 we present the optimal regret of DAL.

**Corollary 4.9.** *Assume that the conditions of Theorem 4.8 hold. In PS-PBs, select $\mathcal{A}_\mathrm{e}$ as in Proposition 4.2. In PS-KBs, select $\mathcal{A}_\mathrm{e}$ as in Proposition 4.3 with $\delta_T := \frac{R d^{1/2 - 2p/d}}{2(C \gamma_T^{2q})^{1/d}}$ for some $C > 0$. In PS-CBs, set $\mathcal{A}_\mathrm{e}$ as in Remark 4.4. Then, DAL attains*

$$R_T = \tilde{\mathcal{O}}(d^p \gamma_T^q (|\mathcal{A}| \log |\Pi|)^r \sqrt{N_T T}).$$

*If the base stationary algorithm has order-optimal regret, DAL retains optimality in PS-PBs, PS-KBs and PS-CBs. This also holds when $N_\mathrm{e} < d$ or $|\mathcal{A}| < d$ in PS-PBs, when $N_\mathrm{e} < \gamma_T$ or $|\mathcal{A}| < \gamma_T$ in PS-KBs, and when $\Pi$ is the universal set of all mappings from $\mathcal{C}$ to $\mathcal{A}$.*

**State-of-the-art Regret.** In line with the black-box philosophy, Corollary 4.9 enables regret bounds across all settings considered, with flexibility in the choice of stationary bandit algorithms. When using specific stationary algorithms from Section 3, which attain the optimal stationary regret mentioned above, DAL matches the state-of-the-art regret bounds in PS-LBs and PS-GLBs at $\tilde{\mathcal{O}}(d\sqrt{N_T T})$. In PS-CBs, DAL achieves the state-of-the-art regret bound of $\tilde{\mathcal{O}}(\sqrt{|\mathcal{A}|N_T T \log |\Pi|})$. More notably, DAL improves the best known bounds in the PS-SCB and PS-KB settings. For PS-SCBs, the strongest, prior-based, bound is due to WeightUCB, which achieves $\tilde{\mathcal{O}}(d^{2/3} T^{2/3} N_T^{1/3})$.[5] DAL improves this to $\tilde{\mathcal{O}}(d\sqrt{N_T T})$ with our algorithmic choices. Although this matches the bound in Russac et al. (2021), their analysis relies on substantially stronger assumptions than ours. For PS-KBs, the prior-free ADA-OPKB achieves $\tilde{\mathcal{O}}(\sqrt{d\gamma_T N_T T})$, while DAL improves this to $\tilde{\mathcal{O}}(\sqrt{\gamma_T N_T T})$. This highlights the interesting feature of DAL: the order-wise dependence on problem parameters from the stationary setting seamlessly transfers to the PS setting without degradation. A detailed comparison of regret bounds is given in the Appendix.

## 4.3 On Drifting Environments

Based on the previous section, at first glance, one can expect that DAL is not able to handle drifting non-stationarity. Our results in Section 3 naturally lead us to ask when and why DAL performs well in drifting environments.

---

[5]While MASTER may be extendable to PS-SCBs, no regret bound is currently known.

As a first step to study this, we perform another experiment with LBs. Specifically, the parameter $\theta_t$ in each time-step $t$ evolves randomly as follows,

$$\theta_{t+1} := \theta_t + \zeta_{t+1}$$

where $\zeta_{t+1} \in \mathbb{R}^d$ is chosen uniformly over a $\delta$-ball. If the resulting $\theta_{t+1}$ violates the norm-bound $S$, we disregard that choice of $\zeta_{t+1}$ and sample again. We sample $\varepsilon_t \sim \mathcal{N}(0, 0.1)$ at each $t$. We compare the cumulative dynamic regret up to time $T$ of DAL+LinUCB with GLR, and WeightUCB over a range of $\delta$'s in Figure 3. The remaining parameters are chosen to be the same as those in Section 3, with the exception of $d = 5$. The DAL algorithm per-

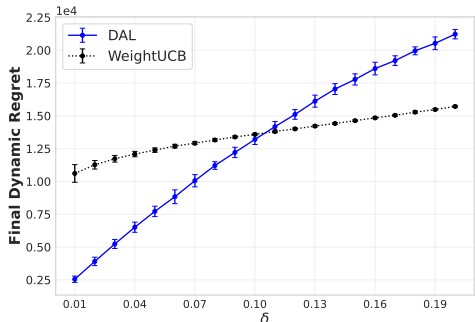

Figure 3: Final dynamic regret vs. radius of change $\delta$: Drifting LBs.

forms better than WeightUCB for smaller values of $\delta$, but the conclusion reverses upon increasing $\delta$. We now shed light into our hypothesis behind the observations from Figure 3. For playing an action $a \in \mathcal{A}$ at time $t + 1$, we get the random reward,

$$X_{t+1} = \langle \theta_t, a \rangle + \langle \zeta_{t+1}, a \rangle + \varepsilon_{t+1}.$$

If the governing parameter does not change, then the reward from playing action $a$ would have been $X'_{t+1} = \langle \theta_t, a \rangle + \varepsilon'_{t+1}$, where $\varepsilon'_{t+1}$ is another realization of the noise. Statistically, a specific instance of $\langle \zeta_{t+1}, a \rangle + \varepsilon_{t+1}$ and $\varepsilon'_{t+1}$ are close to each other, when $\delta$ is small, albeit the resulting (small) mean-shift due to the drift in the governing parameter. For practical purposes, the impact of the drift can be absorbed into the noise term $\varepsilon_{t+1}$ when $\delta$ is small. As a result, one expects an algorithm tailored to handle piecewise stationarity to perform reasonably well for slowly drifting environments. Conversely, if $\delta$ is large, the bias induced by $\zeta_{t+1}$ is large enough to disallow absorbing it into the noise term. Over a few time-steps, the cumulative effect of this compounding bias is then large enough to completely violate the stationarity assumption. With large enough $\delta$, the change in $\theta_t$ over a few time-steps can be considered large enough to trigger a restart.

## 5 SUMMARY AND OUTLOOK

We introduced DAL, a practical, prior-free black-box framework for general PS bandits. Its plug-and-play design integrates seamlessly with a wide range of stationary bandit algorithms and different detectors. Through extensive experiments in both PS and drifting settings—spanning synthetic and real-world benchmarks, DAL consistently outperforms all prior-free baselines, including the black-box gold standard MASTER and the state-of-the-art methods ADA-OPKB and ADA-ILCTB+, and even surpasses leading prior-based methods like WeightUCB and WGP-UCB. Its leading performance in real-world scenarios highlights its value as a practical and effective solution.

On the theoretical side, using existing results and providing novel techniques, we showed that DAL inherits and adapts the regret guarantees of its stationary input algorithm, achieving order-optimal regret under piecewise stationarity, with mild change-point separation. As a result, it matches the best existing bounds in PS-LBs, PS-GLBs and PS-CBs while improving the best known bounds for PS-SCBs and PS-KBs. Regarding drifting non-stationarity, we hypothesized key conditions under which DAL excels–an insight further validated through additional experiments under drifting settings. Our results suggest that a well-designed algorithm for the PS setting can extend to a broad range of drifting scenarios, bridging the gap between these two regimes.

While DAL advances both theory and practice, it opens new directions. First, regret guarantees for detection-based methods in drifting environments remain unexplored. Second, the current regret bounds for DAL rely on a separation condition between change-points—a standard assumption in the detection-based literature (see e.g., (Auer et al., 2019; Besson et al., 2022; Huang et al., 2025))—which nonetheless limits the extent to which DAL achieves fully prior-free theoretical optimality. Addressing these gaps would deepen our understanding of detection-based methods in more continuous forms of non-stationarity. Finally, DAL's modular nature invites extensions to broader settings, including general non-stationary reinforcement learning. We believe that deepening the study of piecewise stationarity may be the key to tackling these broader challenges and DAL can serve as a solid foundation towards that goal.

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

# A EXPERIMENTAL DETAILS

## A.1 ON FORCED EXPLORATION IN FINITE ACTION SPACES

**Covering Set Construction.** In practice, the covering set $\mathcal{A}_e$ is selected according to Propositions 4.2, 4.3, and Remark 4.4 together with the specifications of Corollary 4.9. However, in finite-action settings, the full construction may not be feasible: the action set $\mathcal{A}$ may not contain enough elements to satisfy the required conditions. For instance, in the NS-PB setting, $\mathcal{A}$ may not include $d$ linearly independent actions, while in the NS-KB case, it may lack a full $\delta_T$-covering net for the chosen $\delta_T$ in Corollary 4.9. One might expect that when $|\mathcal{A}_e| < d$ in PS-PBs or $|\mathcal{A}_e| < \gamma_T$ in PS-KBs, the inability to detect all possible changes would degrade DAL's performance. In practice, however, DAL does not need to restart when changes in the reward function leave the mean reward of each action unchanged. Crucially, as discussed in Appendix B.6, DAL retains order-optimality even in these constrained regimes. Accordingly, whenever $|\mathcal{A}| < d$ or $|\mathcal{A}| < \gamma_T$, we simply set $\mathcal{A}_e = \mathcal{A}$. In our experiments, the action set is finite (typically in the hundreds). For PS-PBs, the random generation of actions almost always guarantees $d$ linearly independent vectors. For PS-KBs, since $\gamma_T$ is typically large, we also use the full action set $\mathcal{A}$ as $\mathcal{A}_e$ without impacting performance. On the other hand, since the regret bounds in PS-CBs include $|\mathcal{A}|$, as it is finite, in any PS-CB setting we can simply set $\mathcal{A}_e = \mathcal{A}$.

**Practical Implementations.** For NS-PBs, we construct $\mathcal{A}_e$ by greedily selecting linearly independent actions until we obtain $d$ such vectors, where $d$ is the dimension of the action space. In the NS-KB setting, $\mathcal{A}_e$ is formed by building a $\delta_T$-cover over the bounded action space and choosing the centers of the covering balls. If the action space is continuous and bounded, these centers suffice to cover the space. If the action space is finite and $N_e < d^{2p}\gamma_T^{2q}$, then the entire set $\mathcal{A}$ serves as the covering set, as established in Corollary 4.9. Otherwise, if $N_e > d^{2p}\gamma_T^{2q}$, we select the $d^{2p}\gamma_T^{2q}$ actions closest to the covering-ball centers. Finally, in the NS-CB setting, selecting a smaller $\mathcal{A}_e$ compared to $\mathcal{A}$ does not affect regret, but improves practical performance due to less forced exploration. Thus, depending on the reward function and action set structures, it is recommended to decrease the cardinality of $\mathcal{A}_e$ as much as possible.

**Sensitivity of $\mathcal{A}_e$** As shown in Algorithm 1, DAL's forced exploration depends on $N_e$, the cardinality of $\mathcal{A}_e$. Intuitively, a larger $N_e$ increases the exploration burden, since DAL must select more actions to detect changes. In all cases, DAL limits the cardinality of $\mathcal{A}_e$ to the minimum number of actions needed to characterize the reward function for detection and learning. These cardinalities match the quantities appearing in the minimax stationary regret bounds (e.g., $d$ for PBs, $\gamma_T$ for KBs, and $|\mathcal{A}|$ for CBs). This principle guided our design of the covering-set selection procedures.

- **NS-PBs.** In the NS-PB setting, Proposition 4.2 shows that the cardinality of a suitable covering set is at most $d$. Thus, even if the underlying action space is infinite, DAL only needs to explore at most $d$ actions in $\mathcal{A}_e$. In this sense, DAL is not sensitive to the size of the continuous action space: it pays only a $d$-dependent cost. If there are multiple choices of $d$ linearly independent actions, the practical performance depends on the induced change magnitude $\Delta_c$ (as discussed in Section 4.1). For a fixed non-stationarity model, some choices of $d$ actions may yield larger $\Delta_c$, improving pre- and post-change sample complexity. However, our regret analysis accounts for the worst case over $\Delta_c$, so, at the theoretical level, DAL is not sensitive to which particular $d$ actions are chosen.

- **NS-KBs.** In the NS-KB setting, the sensitivity of DAL is governed by the smoothness of the RKHS. If the Lipschitz constant $BL_u$ is small, the RKHS contains smooth functions, so we can use a relatively large $\delta_T$, leading to a smaller covering set $\mathcal{A}_e$ (and thus a smaller $N_e$). If the RKHS contains less smooth functions (larger $BL_u$), we require a smaller $\delta_T$ to detect changes reliably, which increases $N_e$. Nevertheless, to attain order-optimality DAL only needs to explore at most $\gamma_T$ actions, which is finite and significantly smaller than the (possibly infinite) original action space. As in NS-PBs, DAL is more sensitive to the underlying function class (smoothness) than to the raw size of the continuous action space.

- **NS-CBs.** In the NS-CB case, the action set is finite, and one must fully explore all actions in order to characterize changes in the reward function, since the rewards can be completely uninformative about structural properties beyond their realized values.

**Experimental Choices.** In our experiments, for NS-PBs the action set is sampled from a multi-variate Gaussian distribution, which ensures the existence of $d$ linearly independent actions. Thus, we always set $N_e = d$ using the greedy selection procedure described above. For NS-KBs, the regret bound for $N_e$ obtained from Theorem 4.8 and Corollary 4.9 is extremely large for our horizons, implying that $|\mathcal{A}| < \gamma_T$. Consequently, in all NS-KB experiments we simply take $\mathcal{A}_e = \mathcal{A}$ and set $N_e$ equal to the number of available actions, which yielded optimal performance. Finally, since the reward does not exhibit any structure with the arms in PS-CBs, we simply set $\mathcal{A} = \mathcal{A}_e$.

A.2 REAL-WORLD DATA PREPROCESSING

**Microarchitecture Prefetcher Selection Benchmark.** We introduce a non–stationary bandit dataset derived from the MICRO'23 study of Gerogiannis & Torrellas (2023), built on the SPEC06/17 benchmark suites. Each action corresponds to one of 11 L2 prefetcher configurations (next–line on/off, stream degree, stride degree). The sequence spans $T=26224$ rounds; at round $t$, the reward is the trace–level normalized instructions–per–cycle in $[0, 1]$, computed from performance counters. We obtained the data directly from the original authors, and note that reproducing the exact series from scratch is not feasible without the same stack, microarchitectural parameters, and arm schedules described in the paper. We aim to release the dataset to facilitate real-world experimentation by the bandit research community.

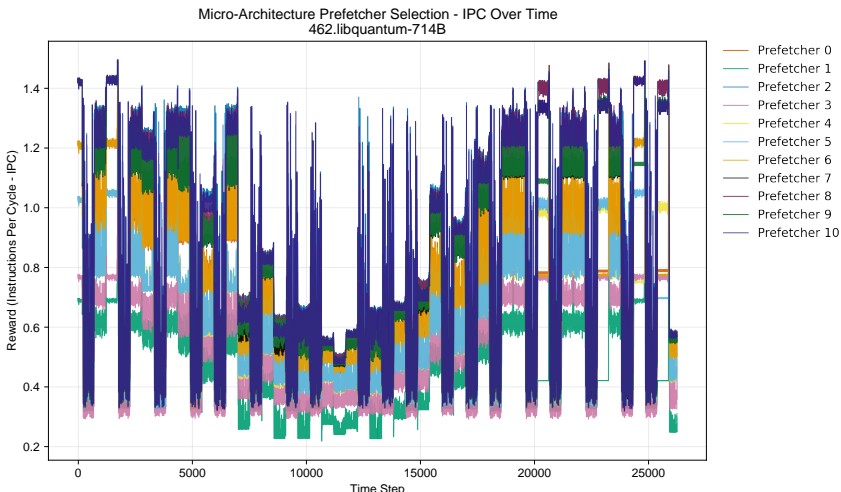

Figure 4: IPC of the prefetchers of the dataset over time.

**Stock Market Data Construction.** Regarding the stock market experiments we follow the procedure of Deng et al. (2022). For the first experiment, we use the data provided in Deng et al. (2022). For the other experiment, we collect daily closing prices of NASDAQ-100 companies using the Yahoo Finance API.[6] We filter out stocks with fewer than $T = 2000$ trading days and align all time series over the most recent $T$ dates. From this pool, we remove stocks with extremely high volatility or mean price to make the problem non-trivial, then select the top $K$ most volatile stocks from the remainder. In both cases, the stock prices are scaled accordingly to lie in $[0, 1]$. Each selected company's scaled closing price series defines the mean-reward sequence for one arm in a $K$-armed bandit problem. Finally, we corrupt the reward at each time step with $\mathcal{N}(0, 0.01)$ noise.

**COVID–NMA Clinical Dataset Construction.** For the clinical benchmark based on the public COVID-NMA pharmacological RCT database (Boutron et al., 2025),[7]. we use only released arm-level counts and metadata and discretize time into calendar months, assigning each trial arm to

---

[6]Data retrieved from Yahoo Finance using the publicly available `yfinance` package. Used solely for non-commercial, academic research purposes.

[7]Data available at: https://doi.org/10.5281/zenodo.14965887

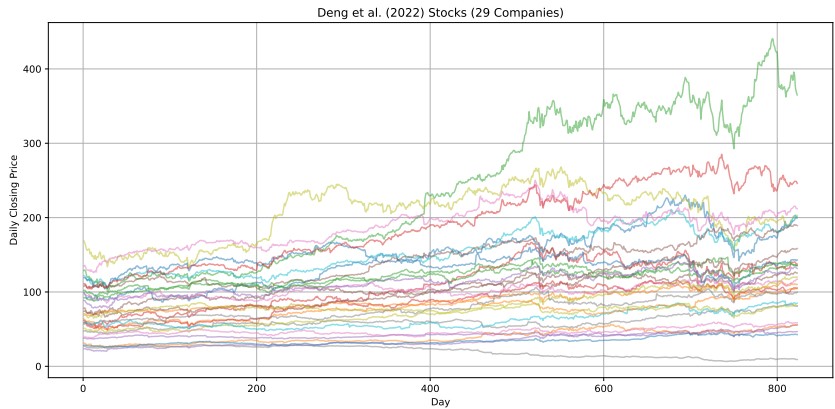

Figure 5: Daily closing prices from the dataset of Deng et al. (2022).

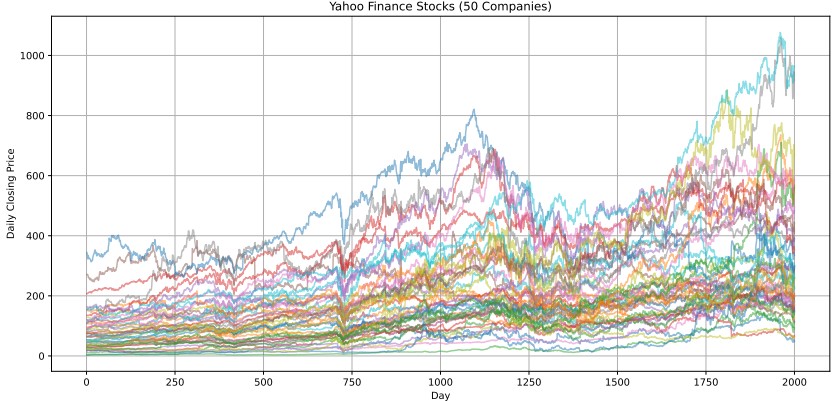

Figure 6: Daily closing prices obtained from Yahoo Finance.

its `Start_date` (falling back to `Pub_date_online`); rows with invalid or missing dates are discarded. We deterministically map case-insensitive rules on treatment type into 13 actions: *Antivirals (any)*, *Anti–inflammatory (steroids/NSAIDs)*, *Interleukin inhibitors*, *Monoclonal antibodies (other)*, *Immunoglobulins/Plasma*, *Antithrombotics*, *Antimicrobials*, *Immunomodulators (non–steroid)*, *Kinase inhibitors*, *Metabolic agents*, *Supportive care*, *Control/Standard care*, and *Other/Unknown*. At the bucket–month level we compute two endpoints: (i) *Clinical Improvement @ D28* (successes = number improved; trials = reported denominator, or baseline $N$ if missing) and (ii) *Survival @ D28* derived from mortality (successes = denominator − deaths). To form a long non–stationary sequence, we adopt a union construction: for each $(k, t, \text{endpoint})$ bin we emit exactly $s_{k,t}$ ones and $n_{k,t}-s_{k,t}$ zeros and concatenate all bins in a fixed order (month, `clinD28`, `mortD28`, bucket). The sequence is fully deterministic; in our run it comprises $T \approx 7.4 \times 10^4$ rounds with 13 actions.

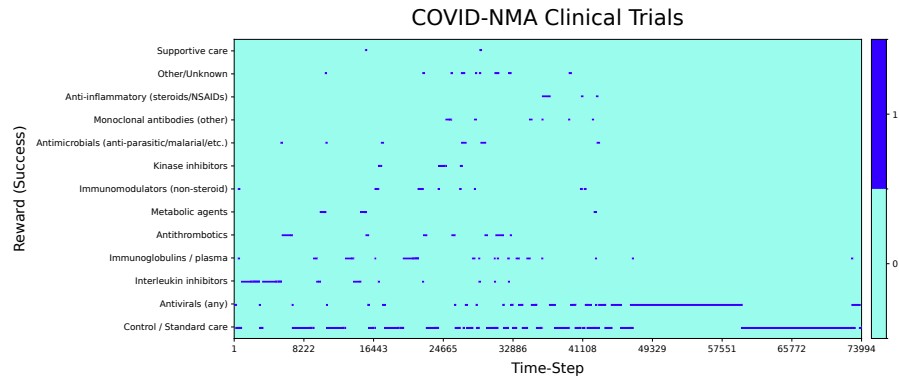

Figure 7: Raw rewards for COVID-NMA Clinical dataset (Boutron et al., 2025).

**Yahoo! R6A Dataset Construction.** For the NS bandit benchmark based on the Yahoo! R6A click log dataset[8], we follow the main procedure provided in Cao et al. (2019a); Zhou et al. (2020). We merge ten consecutive days of logs and we group the data by article ID and compute smoothed click-through rates (CTRs) using centered rolling averages over a 100-round window. This generates a time series of empirical CTRs for each article. We segment the dataset into ten distinct subperiods (each spanning half a day), filtering out actions with missing data or high noise. We further select a set of common actions present in all segments to ensure consistent tracking. We average CTRs within each subperiod and smoothing small deviations below a threshold 0.005. We stack selected actions across multiple days into a single $K \times T$ matrix, where $K$ is the number of valid actions and $T$ the compressed time horizon. To reduce spurious noise and compress the time scale, we apply local smoothing. Finally, we apply post-processing filters to remove (i) globally high-value actions (outliers with inflated CTRs), and (ii) actions that persist as best for too many segments.

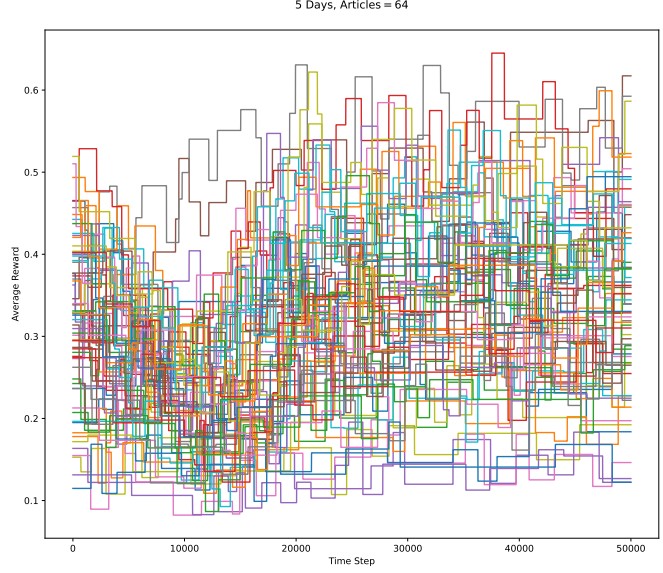

Figure 8: Mean rewards for the Yahoo! R6A dataset.

---

[8]Yahoo! Front Page Today Module User Click Log Dataset: https://webscope.sandbox.yahoo.com.

**Yahoo! R6B Dataset Construction.** We follow a two-stage pipeline tailored to the Yahoo! R6B logs.[8] *Stage 1 (action vocabulary):* we scan the logs to count displays and clicks per article and select the top items using the click-through rate with a minimum display threshold of 2, yielding a fixed action set with mapping $\text{id} \mapsto k \in \{0, \ldots, K-1\}$ with $K = 51$, chosen on the same window as the evaluation files. *Stage 2 (replay log):* we reprocess the files and, for each round $t$, form a feature vector $\mathbf{x}_t$ from the given features, restrict the candidate set to the Top–$K$ vocabulary to obtain $\mathcal{A}_t$, locate the displayed item's index $j_t^\star \in \{0, \ldots, |\mathcal{A}_t| - 1\}$, and record the binary click $X_t \in \{0, 1\}$; we drop rounds where the displayed item lies outside Top–$K$ or $|\mathcal{A}_t| < 2$. To increase coverage at a fixed horizon $T = 50000$, days are merged in a round-robin fashion before truncation. The resulting dataset stores $\{\mathbf{x}_t, \mathcal{A}_t, j_t^\star, r_t, t_t\}_{t=1}^T$. For offline *replay* evaluation, a policy $\pi$ observes $(\mathbf{x}_t, \mathcal{A}_t)$ and proposes $A_t \in \{0, \ldots, |\mathcal{A}_t| - 1\}$; we credit the outcome only when $a_t = j_t^\star$, and report cumulative reward $C_T = \sum_{t=1}^T \mathbb{1}\{a_t = j_t^\star\} r_t$.

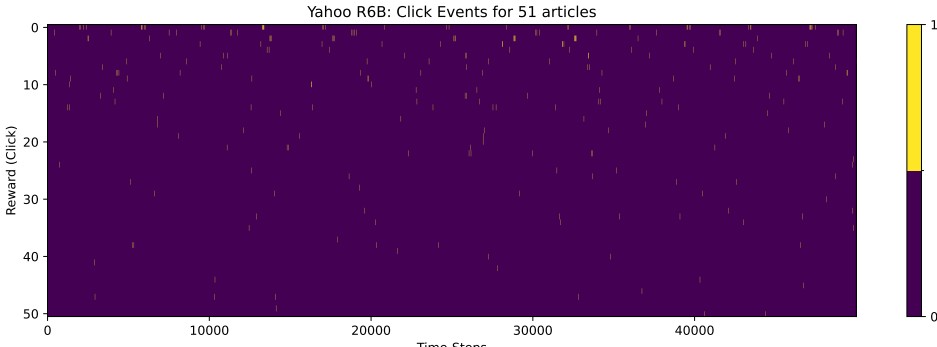

Figure 9: Rewards for the Yahoo! R6B dataset.

**Sensor Correlation Data Construction.** We use the Bioliq dataset from Komiyama et al. (2024), comprising a week of readings from 20 power plant sensors. Following their setup, we construct an NS-SCB environment with 190 actions: the reward is 1 if the last 1000 measurements exceed 2.04, and 0 otherwise. Evaluation is based on cumulative reward. Data available at `https://github.com/edouardfouche/G-NS-MAB/tree/master/data`.

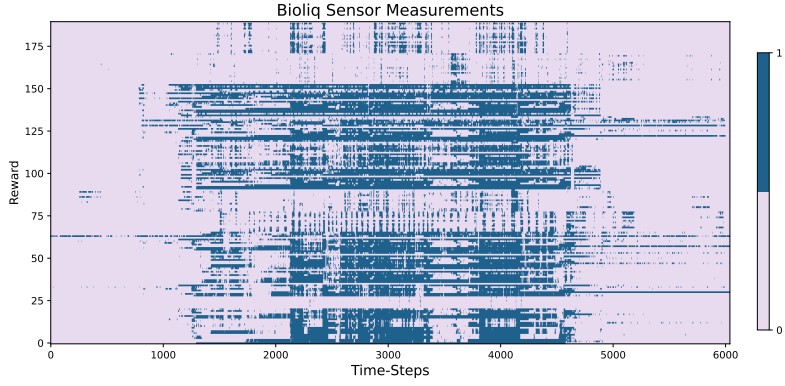

Figure 10: Raw rewards obtained from the Bioliq dataset (Komiyama et al., 2024).

**Ad Recommendation Data Construction.** We evaluate on the Zozo environment, a real-world ad recommender system from Saito et al. (2021), using the preprocessed dataset of Komiyama et al. (2024). We construct an NS-GLB environment with all 80 ads (unlike their 10-action setup), assigning reward 1 to any ad clicked within one second, and 0 otherwise. Evaluation is based on cumulative reward. Data available at `https://github.com/edouardfouche/G-NS-MAB/tree/master/data`.

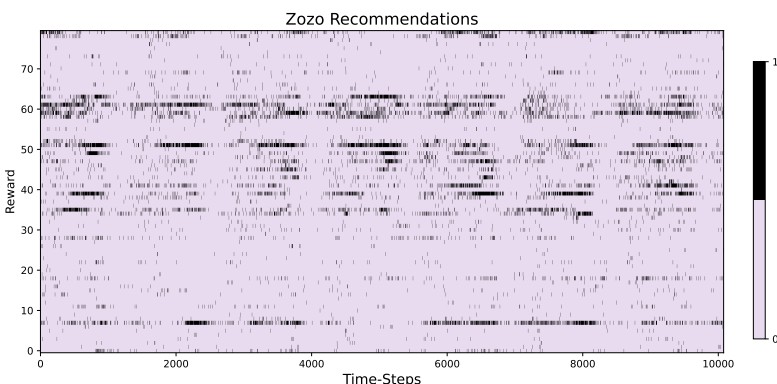

Figure 11: Raw rewards obtained from the Zozo dataset (Komiyama et al., 2024).

**Live Traffic Data Construction.** We construct a NS bandit environment based on the Criteo live traffic dataset (Diemert et al., 2017), following the preprocessing approach of Russac et al. (2019) but modeling the problem as an NS-GLB rather than an NS-LB. Specifically, the dataset includes banners shown to users, associated contextual variables, and whether each banner was clicked. We retain the categorical variables cat1 through cat9, along with campaign, which uniquely identifies each campaign. These categorical features are one-hot encoded, and a dimensionality reduction via Singular Value Decomposition selects 50 resulting features. The parameter vector $\theta^\star$ is estimated using logistic regression. Rewards are then generated from this regression model with added Gaussian noise of variance $\sigma^2 = 0.01$. Unlike Russac et al. (2019), in which the authors employ a single change, we introduce shifts in $\theta^*$ via a geometric change-point model with parameter $\xi = 0.8$, chaging $60\%$ of the $\theta^*$ coordinates at each time-step to $-\theta^*$ and extend the horizon to $T = 50000$.

### A.3 HARDWARE SPECIFICATIONS

All experiments were employed on a desktop using an Intel(R) Xeon(R) W-2245 processor with 32 GB RAM. Each experiment had a total runtime below one hour.

## B THEORETICAL RESULTS

### B.1 GENERAL FORMULATIONS OF GLR AND GSR

For completeness we provide the general mathematical forms of the Generalized Likelihood Ratio (GLR) and the Generalized Shiryaev-Roberts (GSR) tests. Specifically, the GLR test declares a change at time-step $\tau$, such that,

$$\tau := \inf \left\{ n \in \mathbb{N} : G_n \geq \beta\left(n, \delta_{\mathrm{F}}\right) \right\}$$

where the GLR statistics $G_n$ is

$$G_n := \sup_{t \in [n]} \log \left( \frac{\sup_{\theta_0 \in \mathbb{R}} \sup_{\theta_1 \in \mathbb{R}} \prod_{i=1}^{t} f_{\theta_0}\left(X_i\right) \prod_{i=t+1}^{n} f_{\theta_1}\left(X_i\right)}{\sup_{\theta \in \mathbb{R}} \prod_{i=1}^{n} f_{\theta}\left(X_i\right)} \right),$$

while the GSR test declares a change at,

$$\tau := \inf \left\{ n \in \mathbb{N} : \log W_n \geq \beta\left(n, \delta_{\mathrm{F}}\right) + \log n \right\}$$

and the GSR statistic $W_n$ is given by

$$W_n := \frac{1}{n} \sum_{t=1}^{n} \left( \frac{\sup_{\theta_0 \in \mathbb{R}} \sup_{\theta_1 \in \mathbb{R}} \prod_{i=1}^{t} f_{\theta_0}\left(X_i\right) \prod_{i=t+1}^{n} f_{\theta_1}\left(X_i\right)}{\sup_{\theta \in \mathbb{R}} \prod_{i=1}^{n} f_{\theta}\left(X_i\right)} \right).$$

For both cases, $f_\theta$ can be the density of a Gaussian random variable with mean $\theta\sigma^2$ and variance $\sigma^2$ or the density of a Bernoulli random variable with the same mean. Finally, in the general case, we

have that for any false alarm probability $\delta_{\mathrm{F}} \in (0,1)$, the threshold is given by

$$\beta(n, \delta_{\mathrm{F}}) = 6\log(1 + \log(n)) + \frac{5}{2}\log\left(\frac{4n^{3/2}}{\delta_{\mathrm{F}}}\right) + 11.$$

Finally, for the practical implementation of the GLR and GSR in Algorithms 2 and 3, as per Besson et al. (2022); Huang et al. (2025) we have that, for any $n \in \mathbb{N}$ and any $t \in \{1, \ldots, n\}$:

$$\log\left(\frac{\sup_{\theta_0 \in \mathbb{R}} \prod_{i=1}^{t} f_{\theta_0}(X_i) \sup_{\theta_1 \in \mathbb{R}} \prod_{i=t+1}^{n} f_{\theta_1}(X_i)}{\sup_{\theta \in \mathbb{R}} \prod_{i=1}^{n} f_\theta(X_i)}\right)$$
$$= t\,\mathrm{kl}\left(\hat{\mu}_{1:t}; \hat{\mu}_{1:n}\right) + (n - t)\,\mathrm{kl}\left(\hat{\mu}_{t+1:n}; \hat{\mu}_{1:n}\right)$$

where $\hat{\mu}_{t_1:t_2}$ denotes the empirical mean of the reward samples from sample $X_{t_1}$ to sample $X_{t_2}$ with $t_1 < t_2$ and $\mathrm{kl}(x; y)$ is KL-divergence between two Gaussian or Bernoulli distributions, depending on the rewards.

## B.2 REGRET BOUNDS OF DAL IN PIECEWISE STATIONARY ENVIRONMENTS

As discussed in Section 4.2 of the paper, using Corollary 4.9, we can select different algorithms as input for DAL to attain or improve the state-of-the-art regret bounds in PS environments. Combining DAL with different bandit algorithms leads to the results in Table 1. It is evident that DAL matches the state-of-the-art regret bounds in PS-LBs, PS-GLBs and PS-CBs, and DAL improves the best known bounds in the PS-SCB and PS-KB settings. Note that for PS-SCBs, the strongest result corresponds to the prior-based WeightUCB Wang et al. (2023). As demonstrated in the final columns of the table, the order-wise dependence on problem parameters from the stationary setting seamlessly transfers to the PS setting without degradation.

Table 1: Regret bound comparison of algorithms for PS bandits, under the Assumption 4.6. "†" denotes settings with finite number of actions, while MASTER, ADA-OPKB and SCB-WeightUCB also recover the appropriate bounds in this setting. "•" indicates prior-based algorithms.

| PS Setting | Non-Stationary Algorithm | NS Algorithm Regret Bound in $\tilde{\mathcal{O}}(\cdot)$ | DAL Input Regret Bound in $\tilde{\mathcal{O}}(\cdot)$ |
|---|---|---|---|
| PS-LB | MASTER (Wei & Luo, 2021) + LinUCB | $d\sqrt{TN_T}$ | - |
| | ADA-OPKB (Hong et al., 2023) | $d\sqrt{N_T T}$ | - |
| | DAL (ours) + LinUCB (Abbasi-yadkori et al., 2011) | $d\sqrt{N_T T}$ | $d\sqrt{T}$ |
| | DAL (ours) + LinTS (Agrawal & Goyal, 2013) | $d^{3/2}\sqrt{N_T T}$ | $d^{3/2}\sqrt{T}$ |
| | DAL (ours) + PEGE$^\dagger$ (Lattimore & Szepesvári, 2020) | $\sqrt{dN_T T}$ | $\sqrt{dT}$ |
| PS-GLB | MASTER (Wei & Luo, 2021) + GLM-UCB | $d\sqrt{N_T T}$ | - |
| | DAL (ours) + GLM-UCB (Filippi et al., 2010) | $d\sqrt{N_T T}$ | $d\sqrt{T}$ |
| | DAL (ours) + GLM-TSL (Kveton et al., 2020) | $d^{3/2}\sqrt{N_T T}$ | $d^{3/2}\sqrt{T}$ |
| | DAL (ours) + SupCB-GLM$^\dagger$ (Li et al., 2017) | $\sqrt{dN_T T}$ | $\sqrt{dT}$ |
| PS-SCB | SCB-WeightUCB$^\bullet$ (Wang et al., 2023) | $d^{2/3}T^{2/3}N_T^{1/3}$ | – |
| | DAL (ours) + OFU-ECOLog (Faury et al., 2022) | $d\sqrt{N_T T}$ | $d\sqrt{T}$ |
| | DAL (ours) + OFUL-MLogB (Zhang & Sugiyama, 2023) | $d\sqrt{N_T T}$ | $d\sqrt{T}$ |
| | DAL (ours) + OFUGLB (Lee et al., 2024) | $d\sqrt{N_T T}$ | $d\sqrt{T}$ |
| PS-KB | MASTER (Wei & Luo, 2021) + GPUCB | $\gamma_T\sqrt{N_T T}$ | - |
| | ADA-OPKB (Hong et al., 2023) | $\sqrt{d\gamma_T N_T T}$ | - |
| | DAL (ours) + GPUCB (Chowdhury & Gopalan, 2017) | $\gamma_T\sqrt{N_T T}$ | $\gamma_T\sqrt{T}$ |
| | DAL (ours) + REDS (Salgia et al., 2024) | $\sqrt{\gamma_T N_T T}$ | $\sqrt{\gamma_T T}$ |
| PS-CB | MASTER (Wei & Luo, 2021) + ILTCB | $\sqrt{|\mathcal{A}|N_T T \log|\Pi|}$ | - |
| | ADA-ILTCB+ (Chen et al., 2019) | $\sqrt{|\mathcal{A}|N_T T \log|\Pi|}$ | - |
| | DAL (ours) + ILTCB (Agarwal et al., 2014) | $\sqrt{|\mathcal{A}|N_T T \log|\Pi|}$ | $\sqrt{|\mathcal{A}|T \log|\Pi|}$ |
| | DAL (ours) + SquareCB (Foster & Rakhlin, 2020) | $\sqrt{|\mathcal{A}|N_T T \log|\Pi|}$ | $\sqrt{|\mathcal{A}|T \log|\Pi|}$ |

## B.3 PROOF OF PROPOSITION 4.2

In the NS-PB setting, the reward at time $t$ is given by $f_t(a) = \mu(\langle \theta_t, a \rangle)$ for all $a \in \mathcal{A}$, where $\mu$ is injective and $\theta_t \in \mathbb{R}^d$. To detect any changes in $\theta_t$, it suffices to detect changes in the values $\langle \theta_t, a \rangle$ for a suitable set of actions.

Since $\mu$ is injective, each observation $y_{t,i} = \mu(\langle \theta_t, a_i \rangle)$ can be inverted to recover the inner product:

$$\langle \theta_t, a_i \rangle = \mu^{-1}(y_{t,i}).$$

Hence, observing $y_{t,i}$ is equivalent to observing $\langle \theta_t, a_i \rangle$.

Suppose that $\mathcal{A}_e \subseteq \mathcal{A}$ is the maximal linearly independent subset of $\mathcal{A}$. Then, the vector $\theta_t$ is uniquely determined by the inner products $\langle \theta_t, a \rangle$ for $a \in \mathcal{A}_e$. Therefore, any change in $\theta_t$ results in a detectable change in the vector of observations $(y_{t,i})_{a_i \in \mathcal{A}_e}$, meaning that $\mathcal{A}_e$ can be taken to be any maximal linearly independent subset of $\mathcal{A}$, with $|\mathcal{A}_e| \leq d$.

### B.4 PROOF OF PROPOSITION 4.3

In this subsection, we establish the construction of $\mathcal{A}_e$ in the NS-KB setting. According to Lemma 5 from De Freitas et al. (2012), we have that every $f \in H_k$ with $\|f\|_{H_k} \leq B$ is Lipschitz continuous, satisfying the following,

$$|f(x) - f(y)| \leq B\,L_u\,\|x - y\|_2, \ \forall\, x, y \in \mathcal{A}, \quad \text{where } L_u := \sup_{z \in D} \max_{i,j \leq d} \left[ \frac{\partial^2 k(p,q)}{\partial p_i\, \partial q_j} \right]^{1/2}_{p=q=z}.$$

Recall that $\mathcal{V}_T$ corresponds to the set of centers of the balls of an arbitrary $\delta_T$-cover of $\mathcal{A} \subseteq [0, R]^d$, with $\delta_T = L_T/(2BL_u)$ for some arbitrary $L_T > 0$. Let $[a]_e$ denote the action in $\mathcal{V}_T$ that is the closest to $a \in \mathcal{A}$, i.e., $[a]_e = \operatorname{argmin}_{x \in \mathcal{P}_T} \|a - x\|_2$. Then, we can leverage the Lipschitz property of functions in the RKHS to obtain the following upper bound: For any $a \in \mathcal{A}$ and $f \in H_k$ with $\|f\|_{H_k} \leq B$,

$$|f(a) - f([a]_e)| \overset{(a)}{\leq} BL_u\|a - [a]_e\|_2 \overset{(b)}{\leq} BL_u\delta_T. \tag{1}$$

Step $(a)$ follows from the Lipschitz property in Lemma 5 of De Freitas et al. (2012), and step $(b)$ results from the definition of a $\delta_T$-cover. Then, for any arbitrary functions $f$ and $f'$ in $H_k$ with $\|f\|_{H_k}, \|f'\|_{H_k} \leq B$ and action $\tilde{a} \in \mathcal{A}$, we have

$$|f([\tilde{a}]_e) - f'([\tilde{a}]_e)| \geq |f(\tilde{a}) - f'(\tilde{a})| - |f(\tilde{a}) - f([\tilde{a}]_e)| - |f'(\tilde{a}) - f'([\tilde{a}]_e)|$$
$$\overset{(a)}{\geq} |f(\tilde{a}) - f'(\tilde{a})| - 2BL_u\delta_T = |f(\tilde{a}) - f'(\tilde{a})| - L_T \overset{(b)}{>} 0$$

where step $(a)$ is due to equation 1, and step $(b)$ is due to the assumption in Proposition 4.3. This indicates that the value of the reward function at $[\tilde{a}]_e$ must change by a non-zero amount. Thus, one can use observations from action $[\tilde{a}]_e$ in order to deduce whether the reward function has changed its value in action $\tilde{a}$. In addition, by the upper bound on the covering number, the cardinality of $\mathcal{V}_T$ is upper bounded by $\lceil \sqrt{d}R/2\delta_T \rceil^d = \lceil \sqrt{d}BL_uR/L_T \rceil^d$ .

### B.5 PROOF OF THEOREM 4.8

For PS-PBs and PS-KBs, the proof of Theorem 4.8 follows exactly the same as those of Theorem 1 and Corollary 1 in Huang et al. (2025), with the number of arms replaced by $N_e$, due to the different number of actions in the covering set. For completeness, we provide a proof sketch of Theorem 4.8: First, we partition the regret into two cases. If no false alarm occurs and all changes are detected within a short delay, we can separate the regret into three components: the regret due to forced exploration, the regret during the short detection (restart) delay after changes, and the regret incurred by the stationary bandit algorithm after the change is detected. If not, we use a crude linear bound to bound the regret and show that the probability of false alarm and that of late detection are low, which ensures that the regret due to detection failure is small.

For PS-CBs, the proof of Theorem 4.8 the definition of successful detection events should be modified as follows:

Consider a PS-CB environment satisfying the change-point separation condition in Theorem 4.8, and recall that $\mathcal{D}$ is the change detector of DAL. Let $\tau_k$ be the $k^{\text{th}}$ detection point for $k \in \mathbb{N}$, i.e.,

$$\tau_k := \inf \{t > \tau_{k-1} : \mathcal{D}(H_{c,a}) = \text{Detection at time-step } t \text{ for some } (c, a) \in \mathcal{C} \times \mathcal{A}_e \}, \tag{2}$$

where $\tau_0 = 0$. Recall that $\nu_0 := 1$ and $\nu_{N_T+1} := T + 1$. We define the following events:

$$\mathcal{G}_k := \{\forall\, l \in [k-1],\ \tau_l \in \{\nu_l, \ldots, \nu_l + \ell_l - 1\}\} \cap \{\tau_k > \nu_k\},\ k \in [N_T]. \tag{3}$$

The event $\mathcal{G}_k$ represents the "good event" up to the $k^{\text{th}}$ detection point $\mathcal{G}_k$ in which the first $k$ changes are detected within the latency. For notational convenience, we define $\mathcal{G}_0$ to be the universal space. Then, we have the following:

$$
R_T = \mathbb{E}\left[\sum_{k=1}^{N_T+1}\sum_{t=\nu_{k-1}}^{\nu_k-1}\max_{\pi\in\Pi} f_t(C_t, \pi(C_t)) - f_t(C_t, A_t)\right]
$$

$$
= \sum_{k=1}^{N_T+1}\mathbb{E}\left[\sum_{t=\nu_{k-1}}^{\nu_k-1}\max_{\pi\in\Pi} f_t(C_t, \pi(C_t)) - f_t(C_t, A_t)\right]
$$

$$
= \sum_{k=1}^{N_T+1}\mathbb{P}\left(\mathcal{G}_k^c\right)\mathbb{E}\left[\sum_{t=\nu_{k-1}}^{\nu_k-1}\max_{\pi\in\Pi} f_t(C_t, \pi(C_t)) - f_t(C_t, A_t)\,\middle|\,\mathcal{G}_k^c\right]
$$

$$
+ \sum_{k=1}^{N_T+1}\mathbb{E}\left[\mathbb{1}\left\{\mathcal{G}_k\right\}\sum_{t=\nu_{k-1}}^{\nu_k-1}\max_{\pi\in\Pi} f_t\left(C_t, \pi\left(C_t\right)\right) - f_t\left(C_t, A_t\right)\right]
$$

$$
\overset{(a)}{\leq} \sum_{k=1}^{N_T+1}\bar{\Delta}\left(\nu_k - \nu_{k-1}\right)\mathbb{P}\left(\mathcal{G}_k^c\right) + \sum_{k=1}^{N_T+1}\mathbb{E}\left[\mathbb{1}\left\{\mathcal{G}_k\right\}\sum_{t=\nu_{k-1}}^{\nu_k-1}\max_{\pi\in\Pi} f_t\left(C_t, \pi\left(C_t\right)\right) - f_t\left(C_t, A_t\right)\right]
\tag{4}
$$

where $\bar{\Delta}$ in step $(a)$ is the maximum gap between the mean rewards of two actions, over all contexts, actions, and time-steps, i.e., $\bar{\Delta} := \max_{c\in\mathcal{C}, a\in\mathcal{A}, t\in[T]}\left(\max_{\pi\in\Pi} f_t(c, \pi(c)) - f_t(c, a)\right)$. For convenience in the proof of the upper bound on the probability of bad event $\mathbb{P}\left(\mathcal{G}_k^c\right)$, define

$$\mathcal{E}_k := \{\forall\, l \in [k-1],\ \tau_l \in \{\nu_l, \ldots, \nu_l + \ell_l - 1\}\},\ k \in [N_T]. \tag{5}$$

$\mathbb{P}\left(\mathcal{G}_k^c\right)$ is upper bounded by the following modified union bound, which decomposes the bad event into false alarm events and late detection events:

$$
\mathbb{P}\left(\mathcal{G}_k^c\right) = \mathbb{P}\left(\{\exists\, l \in [k-1],\ \tau_l \notin \{\nu_l, \ldots, \nu_l + \ell_l - 1\}\} \cup \{\tau_k \leq \nu_k\}\right)
$$

$$
= \sum_{l=1}^{k-1}\mathbb{P}\left(\tau_l \notin \{\nu_s, \ldots, \nu_l + \ell_l - 1\},\ \mathcal{E}_{l-1}\right) + \mathbb{P}\left(\tau_k \leq \nu_k,\ \mathcal{E}_{k-1}\right)
$$

$$
= \sum_{l=1}^{k-1}\mathbb{P}\left(\mathcal{E}_{l-1}\right)\mathbb{P}\left(\tau_l \notin \{\nu_l, \ldots, \nu_l + \ell_l - 1\}\,\middle|\,\mathcal{E}_{l-1}\right) + \mathbb{P}\left(\mathcal{E}_{k-1}\right)\mathbb{P}\left(\tau_k \leq \nu_k\,\middle|\,\mathcal{E}_{k-1}\right)
$$

$$
\overset{(a)}{\leq} \sum_{l=1}^{k-1}\mathbb{P}\left(\tau_l \notin \{\nu_l, \ldots, \nu_l + \ell_l - 1\}\,\middle|\,\mathcal{E}_{l-1}\right) + \mathbb{P}\left(\tau_k \leq \nu_k\,\middle|\,\mathcal{E}_{k-1}\right)
$$

$$
= \sum_{l=1}^{k}\underbrace{\mathbb{P}\left(\tau_l < \nu_l\,\middle|\,\mathcal{E}_{l-1}\right)}_{\Phi_1} + \sum_{l=1}^{k-1}\underbrace{\mathbb{P}\left(\tau_l \geq \nu_l + \ell_l\,\middle|\,\mathcal{E}_{l-1}\right)}_{\Phi_2}
\tag{6}
$$

where $(a)$ is due to the fact that $\mathbb{P}\left\{\mathcal{E}_{k-1}\right\} \leq 1$. We then separately bound $\Phi_1$ and $\Phi_2$.

• *Upper-Bounding $\Phi_1$:* Recall that $\mathcal{A}_e = \left\{a^{(i)}, i \in [N_e]\right\}$ is the covering set, and that $H_{(c,a^{(i)})}$ is the change detector history list associated with the context-action pair $\left(c, a^{(i)}\right)$. For any context $c \in \mathcal{C}$, $i \in [N_e]$, and $u \in \mathbb{N}$, we define $t'_{(c,i),u}$ to be the $u^{\text{th}}$ time-step after $\tau_{l-1}$ at which $C_t = c$ and $(t - \tau_{l-1} - 1)\ \mathrm{mod}\ \lceil N_e/\alpha_l\rceil = i - 1$, i.e.,

$$t'_{(c,i),u} := \inf\left\{t > t'_{(c,i),u-1} : C_t = c,\ (t - \tau_{l-1} - 1)\ \ \mathrm{mod}\ \left\lceil\frac{N_e}{\alpha_l}\right\rceil = i - 1\right\} \tag{7}$$

where $t'_{(c,i),0} = \tau_{l-1}$. Then, we define $n_{c,i}(t)$ to be the number of time-steps between $\tau_{l-1} + 1$ and $t$ at which $C_t = c$ and $(t - \tau_{l-1} - 1) \mod \lceil N_e/\alpha_l \rceil = i - 1$, which is the number of samples obtained due to force exploration and added in the history $H_{(c,a^{(i)})}$ if there are no restarts after $\tau_{l-1}$, i.e.,

$$n_{(c,i)}(t) := \sum_{s=\tau_{l-1}+1}^{t} \mathbb{1}\left\{C_t = c, (t - \tau_{l-1} - 1) \mod \left\lceil \frac{N_e}{\alpha_l} \right\rceil = i - 1\right\}. \tag{8}$$

We also use $\tau_{(c,i)}$ to denote the stopping time of the change detector associated with arm $a^{(i)} \in \mathcal{A}_e$ after the $(l-1)^{\text{th}}$ detection point $\tau_{l-1}$, i.e.,

$$\tau_{(c,i)} := \inf\left\{u \in \mathbb{N} : \mathcal{D}\left(H_{c,a^{(i)}}\right) = \text{Detection at time-step } t'_{(c,i),u}\right\}. \tag{9}$$

The stopping time $\tau_{(c,i)}$ operates independently from other stopping times, and does not stop if other stopping times get triggered earlier. Let $\mathbb{P}_\infty$ denote the probability measure at which $f_t = f_{\nu_l}$ for all $t > \nu_l$, i.e., the probability measure under which the CB becomes stationary after the $k^{\text{th}}$ change-point. Then, for all $l \in [N_T + 1]$, we have

$$\mathbb{P}\left(\tau_l < \nu_l \middle| \mathcal{E}_{l-1}\right) = \mathbb{P}\left(\exists \left(c, a^{(i)}\right) \in \mathcal{C} \times \mathcal{A}_e : \tau_{(c,i)} \in \left[n_{(c,i)}(\nu_l - 1)\right] \middle| \mathcal{E}_{l-1}\right)$$

$$\overset{(a)}{\leq} \sum_{c \in \mathcal{C}} \sum_{i=1}^{N_e} \mathbb{P}\left(\tau_{(c,i)} \in \left[n_{(c,i)}(\nu_l - 1)\right] \middle| \mathcal{E}_{l-1}\right)$$

$$\overset{(b)}{\leq} \sum_{c \in \mathcal{C}} \sum_{i=1}^{N_e} \mathbb{P}\left(\tau_{(c,i)} \leq T \middle| \mathcal{E}_{l-1}\right) \tag{10}$$

$$\overset{(c)}{\leq} \sum_{c \in \mathcal{C}} \sum_{i=1}^{N_e} \delta_F$$

$$= |\mathcal{C}| N_e \delta_F$$

where step $(a)$ results from a union bound. Due to the fact that the rewards between $\tau_{l-1}$ and $\nu_l$ are i.i.d. across time-steps and actions given the past event $\mathcal{E}_{l-1}$ (as there are no changes between $\tau_{l-1}$ and $\nu_l$), we can change the measure to $\mathbb{P}_\infty$ in step $(b)$. In addition, because $[n_a(\nu_l - 1)] \subseteq [T]$, the event $\{\tau_{a,l} \in [n_a(\nu_l - 1)]\} \subseteq \{\tau_{a,l} \leq T\}$. In step $(c)$, since the reward samples $\left\{X_{t'_{(c,i),u}}\right\}_{u \geq 1}$ are i.i.d. sub-Gaussian for each $(c,i) \in \mathcal{C} \times [N_e]$, we can apply the false alarm probability upper bound for the GLR and GSR tests in Huang & Veeravalli (2025) (see Section 4.1).

• *Upper Bounding* $\Phi_2$: Let $(c^*, i^*)$ be the context-action pair at which the mean reward function changes the most at $\nu_l$, i.e.,

$$(c^*, i^*) = \underset{c \in \mathcal{C}, i \in [N_e]}{\arg\max} \left|f_{\nu_l}\left(c, a^{(i)}\right) - f_{\nu_l - 1}\left(c, a^{(i)}\right)\right|. \tag{11}$$

We define the events $\mathcal{M}_l$ and $\mathcal{L}_l$ as follows:

$$\mathcal{M}_l := \left\{\sum_{t=\tau_{l-1}+1}^{\nu_l - 1} \mathbb{1}\left\{C_t = c^*, (t - \tau_{l-1} - 1) \mod \left\lceil \frac{N_e}{\alpha_l} \right\rceil = i^* - 1\right\} \geq m_{\mathcal{D}}\right\}, \tag{12}$$

$$\mathcal{L}_l := \left\{\sum_{t=\nu_l}^{\nu_l + \ell_l - 1} \mathbb{1}\left\{C_t = c^*, (t - \tau_{l-1} - 1) \mod \left\lceil \frac{N_e}{\alpha_l} \right\rceil = i^* - 1\right\} \geq \ell_{\mathcal{D}}\right\}. \tag{13}$$

When $\tau_l \geq \nu_l + \ell_l$, there are at least $m_{\mathcal{D}}$ reward samples following $f_{\nu_l - 1}$ in $H_{(c^*, a^{(i^*)})}$ under the event $\mathcal{M}_l$, and there are at least $\ell_{\mathcal{D}}$ reward samples following $f_{\nu_l}$ in $H_{(c^*, a^{(i^*)})}$ under the event $\mathcal{L}_l$. Then, we have,

$$\mathbb{P}\left(\tau_l \geq \nu_l + \ell_l \middle| \mathcal{E}_{l-1}\right)$$

$$\leq \mathbb{P}\left(\{\tau_l \geq \nu_l + \ell_l\} \cup \mathcal{M}_l^c \cup \mathcal{L}_l^c \middle| \mathcal{E}_{l-1}\right)$$

$$= \mathbb{P}\left(\mathcal{M}_l^c \cup \mathcal{L}_l^c \middle| \mathcal{E}_{l-1}\right) + \mathbb{P}\left(\{\tau_l \geq \nu_l + \ell_l\} \cap \mathcal{M}_l \cap \mathcal{L}_l \middle| \mathcal{E}_{l-1}\right)$$

$$= \mathbb{P}\left(\mathcal{M}_l^c \cup \mathcal{L}_l^c \middle| \mathcal{E}_{l-1}\right) + \mathbb{P}\left(\mathcal{M}_l \cap \mathcal{L}_l \middle| \mathcal{E}_{l-1}\right) \mathbb{P}\left(\tau_l \geq \nu_l + \ell_l \middle| \mathcal{M}_l \cap \mathcal{L}_l \cap \mathcal{E}_{l-1}\right)$$

$$\overset{(a)}{\leq} \mathbb{P}\left(\mathcal{M}_l^c \middle| \mathcal{E}_{l-1}\right) + \mathbb{P}\left(\mathcal{L}_l^c \middle| \mathcal{E}_{l-1}\right) + \mathbb{P}\left(\tau_l \geq \nu_l + \ell_l \middle| \mathcal{M}_l \cap \mathcal{L}_l \cap \mathcal{E}_{l-1}\right) \tag{14}$$

where step $(a)$ follows from a union bound and the fact that $\mathbb{P}\left(\mathcal{M}_l \cap \mathcal{L}_l | \mathcal{E}_{l-1}\right) \leq 1$. For upper bounding the first two terms, we use the fact that that given $\mathcal{E}_{l-1}$, for any $i \in [N_e]$ and $u, v > \tau_{l-1}$ where $v < u$,

$$\sum_{t=v+1}^{u} \mathbb{1}\left\{(t - \tau_k - 1) \mod \left\lceil \frac{N_e}{\alpha_l} \right\rceil = i - 1\right\} \geq \left\lfloor \frac{u - v}{\lceil N_e/\alpha_l \rceil} \right\rfloor. \tag{15}$$

The inequality in equation 15 holds with equality when $u - v$ is divisible by $\lceil N_e/\alpha_l \rceil$. Recall that $n_{(c,i)}(t)$ is the number of time-steps between $\tau_{l-1} + 1$ and $t$ at which $C_t = c$ and $(t - \tau_{l-1} - 1 \mod \lceil N_e/\alpha_l \rceil) = i - 1$ (see equation 8). Then, we have

$$\mathbb{E}\left[n_{(c^*, i^*)}(\nu_l - 1) - n_{(c^*, i^*)}(\tau_{l-1}) | \mathcal{E}_{l-1}\right]$$

$$\overset{(a)}{\geq} \mathbb{E}\left[n_{(c^*, i^*)}(\nu_l - 1) - n_{(c^*, i^*)}(\nu_l - m_l - 1) | \mathcal{E}_{l-1}\right]$$

$$= \mathbb{E}\left[\sum_{t=\nu_l - m_l}^{\nu_l - 1} \mathbb{1}\left\{C_t = c^*, (t - \tau_k - 1) \mod \left\lceil \frac{N_e}{\alpha_l} \right\rceil = i^* - 1\right\} \Bigg| \mathcal{E}_{l-1}\right]$$

$$= \sum_{t=\nu_l - m_l}^{\nu_l - 1} \mathbb{P}(C_t = c^* | \mathcal{E}_{l-1}) \mathbb{1}\left\{(t - \tau_k - 1) \mod \left\lceil \frac{N_e}{\alpha_l} \right\rceil = i^* - 1\right\}$$

$$\overset{(b)}{=} \sum_{t=\nu_l - m_l}^{\nu_l - 1} \mathcal{P}_t(c) \mathbb{1}\left\{(t - \tau_k - 1) \mod \left\lceil \frac{N_e}{\alpha_l} \right\rceil = i - 1\right\}$$

$$\overset{(c)}{\geq} s \sum_{t=\nu_l - m_l}^{\nu_l - 1} \mathbb{1}\left\{(t - \tau_k - 1) \mod \left\lceil \frac{N_e}{\alpha_l} \right\rceil = i - 1\right\}$$

$$\overset{(d)}{=} s \left\lfloor \frac{m_l}{\lceil N_e/\alpha_l \rceil} \right\rfloor$$

$$= s \left\lceil \frac{m_{\mathcal{D}}}{s} + \frac{\log T}{4s^2} + \sqrt{\frac{m_D \log T}{2s^3} + \frac{(\log T)^2}{16s^4}} \right\rceil, \tag{16}$$

and

$$\mathbb{E}\left[n_{(c^*, i^*)}(\nu_l + \ell_l - 1) - n_{(c^*, i^*)}(\nu_l - 1)\right]$$

$$= \mathbb{E}\left[\sum_{t=\nu_l}^{\nu_l + \ell_l - 1} \mathbb{1}\left\{C_t = c^*, (t - \tau_k - 1) \mod \left\lceil \frac{N_e}{\alpha_l} \right\rceil = i^* - 1\right\}\right]$$

$$= \sum_{t=\nu_l}^{\nu_l + \ell_l - 1} \mathbb{P}(C_t = c^* | \mathcal{E}_{l-1}) \mathbb{1}\left\{(t - \tau_k - 1) \mod \left\lceil \frac{N_e}{\alpha_l} \right\rceil = i^* - 1\right\}$$

$$\overset{(e)}{=} \sum_{t=\nu_l}^{\nu_l + \ell_l - 1} \mathcal{P}_t(c) \mathbb{1}\left\{(t - \tau_k - 1) \mod \left\lceil \frac{N_e}{\alpha_l} \right\rceil = i - 1\right\}$$

$$\overset{(f)}{\geq} s \sum_{t=\nu_l}^{\nu_l + \ell_l - 1} \mathbb{1}\left\{(t - \tau_k - 1) \mod \left\lceil \frac{N_e}{\alpha_l} \right\rceil = i - 1\right\}$$

$$\overset{(g)}{=} s \left\lfloor \frac{\ell_l}{\lceil N_e/\alpha_l \rceil} \right\rfloor$$

$$= s \left\lceil \frac{\ell_{\mathcal{D}}}{s} + \frac{\log T}{4s^2} + \sqrt{\frac{\ell_D \log T}{2s^3} + \frac{(\log T)^2}{16s^4}} \right\rceil. \tag{17}$$

In step $(a)$, since $\tau_{l-1} \leq \nu_{l-1} + \ell_{l-1} - 1$ given $\mathcal{E}_{l-1}$ and $\nu_l - \nu_{l-1} \geq \ell_{l-1} + m_l$ by Assumption 4.6, $\tau_{l-1} \leq \nu_l - m_l - 1$ and thus $n_{(c^*, i^*)}(\nu_l - 1) \leq n_{(c^*, i^*)}(\nu_l - m_l - 1)$. Steps $(b)$ and $(e)$ follow from the independence between $(C_t)_{t > \tau_l}$ and $\mathcal{E}_{l-1}$. Steps $(c)$ and $(f)$ stem from the definition of $s$

in Theorem 4.6 ($s = \min_{c \in \mathcal{C}, t \in [T]: \mathcal{P}_t(c) > 0} \mathcal{P}_t(c)$). Steps $(d)$ and $(g)$ result from equation 15, as $m_l$ and $\ell_l$ are divisible by $\lceil N_{\mathrm{e}}/\alpha_l \rceil$. Therefore,

$$\mathbb{P}\left(\mathcal{M}_l^c \big| \mathcal{E}_{l-1}\right)$$

$$= \mathbb{P}\left(\sum_{t=\tau_l+1 : (t-\tau_k-1) \bmod \lceil N_{\mathrm{e}}/\alpha_l \rceil = i^*-1}^{\nu_l-1} \mathbb{1}\left\{C_t = c^*\right\} \le m_D \bigg| \mathcal{E}_{l-1}\right)$$

$$\overset{(a)}{\le} \exp\left(\frac{-2\left(\mathbb{E}\left[n_{(c^*,i^*)}(\nu_l-1) - n_{(c^*,i^*)}(\tau_{l-1})\right] - m_{\mathcal{D}}\right)^2}{\sum_{t=\tau_l+1}^{\nu_l-1} \mathbb{1}\left\{(t-\tau_k-1) \bmod \lceil N_{\mathrm{e}}/\alpha_l \rceil = i^*-1\right\}}\right)$$

$$\overset{(b)}{\le} \exp\left(\frac{-2\left(s\left\lceil \ell_D/s + \log(T)/4s^2 + \sqrt{\ell_D \log T/2s^3 + (\log T)^2/16s^4}\right\rceil - \ell_{\mathcal{D}}\right)^2}{\left\lceil \ell_D/s + \log(T)/4s^2 + \sqrt{\ell_D \log T/2s^3 + (\log T)^2/16s^4}\right\rceil}\right)$$

$$\le T^{-1}, \tag{18}$$

and

$$\mathbb{P}\left(\mathcal{L}_l^c \big| \mathcal{E}_{l-1}\right)$$

$$= \mathbb{P}\left(\sum_{t=\nu_l : (t-\tau_k-1) \bmod \lceil N_{\mathrm{e}}/\alpha_l \rceil = i^*-1}^{\nu_l+\ell_l-1} \mathbb{1}\left\{C_t = c^*\right\} \le \ell_D \bigg| \mathcal{E}_{l-1}\right)$$

$$\overset{(c)}{\le} \exp\left(\frac{-2\left(\mathbb{E}\left[n_{(c^*,i^*)}(\nu_l+\ell_l-1) - n_{(c^*,i^*)}(\nu_l-1)\right] - \ell_{\mathcal{D}}\right)^2}{\sum_{t=\nu_l}^{\nu_l+\ell_l-1} \mathbb{1}\left\{(t-\tau_k-1) \bmod \lceil N_{\mathrm{e}}/\alpha_l \rceil = i^*-1\right\}}\right)$$

$$\overset{(d)}{\le} \exp\left(\frac{-2\left(s\left\lceil \ell_D/s + \log(T)/4s^2 + \sqrt{\ell_D \log T/2s^3 + (\log T)^2/16s^4}\right\rceil - \ell_{\mathcal{D}}\right)^2}{\left\lceil \ell_D/s + \log(T)/4s^2 + \sqrt{\ell_D \log T/2s^3 + (\log T)^2/16s^4}\right\rceil}\right)$$

$$\le T^{-1}. \tag{19}$$

In steps $(a)$ and $(c)$, we apply Hoeffding's inequality, as $\{\mathbb{1}\{C_t = c^*\}\}_{t \ge \tau_l}$ is a sequence of i.i.d. Bernoulli random variables with parameter $\mathcal{P}_t(c)$. In steps $(b)$ and $(d)$, we apply equation 17.

Before bounding the third term in equation 14, recall the definition of the stopping time of the change detector associated with arm $a^{(i)}$ after the $(l-1)^{\text{th}}$ detection point in equation 9. Without loss of generality, we assume that $\nu_l \le T - \ell_l$; otherwise, there is no need to detect the change because the horizon will end soon after the change occurs. We can derive that

$$\mathbb{P}\left(\tau_l \ge \nu_l + \ell_l | \mathcal{E}_{l-1} \cap \mathcal{M}_l \cap \mathcal{L}_l\right)$$

$$= \mathbb{P}\left(\forall (c,i) \in \mathcal{C} \times [N_{\mathrm{e}}], \; \tau_{(c,i)} > n_{(c,i)}(\nu_l+\ell_l-1) \big| \mathcal{E}_{l-1} \cap \mathcal{M}_l \cap \mathcal{L}_l\right)$$

$$\overset{(a)}{\le} \mathbb{P}\left(\tau_{(c^*,i^*)} > n_{(c^*,i^*)}(\nu_l+\ell_l-1) \big| \mathcal{E}_{l-1} \cap \mathcal{M}_l \cap \mathcal{L}_l\right)$$

$$\overset{(b)}{\le} \mathbb{P}\left(\tau_{(c^*,i^*)} > n_{(c^*,i^*)}(\nu_l-1) + \ell_{\mathcal{D}} \big| \mathcal{E}_{l-1} \cap \mathcal{M}_l \cap \mathcal{L}_l\right)$$

$$\overset{(c)}{\le} \sup_{\nu \in \{m_{\mathcal{D}}+1,\ldots,T-\ell_{\mathcal{D}}\}} \mathbb{P}\left(\tau_{(c^*,i^*)} \ge \nu + \ell_{\mathcal{D}} \big| \mathcal{E}_{l-1} \cap \mathcal{M}_l \cap \mathcal{L}_l\right)$$

$$\overset{(d)}{\le} \delta_{\mathrm{D}} \tag{20}$$

where step $(a)$ comes from the fact that $\{(c^*, i^*)\} \subseteq \mathcal{C} \times [N_{\mathrm{e}}]$, and step $(b)$ stems from the fact that $n_{(c^*,i^*)}(\nu_l+\ell_l-1) - n_{(c^*,i^*)}(\nu_l-1) \ge \ell_{\mathcal{D}}$ given $\mathcal{L}_l$. Step $(c)$ results from the fact that $n_{(c^*,i^*)}(\nu_l-1) \ge m_{\mathcal{D}}$ given $\mathcal{M}_l$ and $\nu_l \le T - \ell_l$. Recall the definition of $t'_{(c,i),u}$ in equation 7. Step $(d)$ follows from the definition of latency in Section 4.1, as the reward sequence $\left\{X_{t'_{(c^*,i^*),u}}\right\}_{u \ge 1}$ are independent sub-Gaussian whose distribution changes at $\nu$, given $\mathcal{E}_{l-1}$ and the context sequence $\{C_t\}_{t \ge 1}$. Plugging equation 18, equation 19, and equation 20 into equation 14, we have

$$\mathbb{P}\left(\tau_l \ge \nu_l + \ell_l | \mathcal{E}_{l-1}\right) \le 2T^{-1} + \delta_{\mathrm{D}}. \tag{21}$$

This completes bounding $\Phi_1$ and $\Phi_2$. Plugging equation 10 and equation 20 into equation 6, we obtain

$$\mathbb{P}\{\mathcal{G}_k^c\} \le |\mathcal{C}|N_e k \delta_{\mathrm{F}} + (k-1)\left(2T^{-1} + \delta_{\mathrm{D}}\right). \tag{22}$$

This bounds the first term in equation 4.

For convenience in bounding the second term in equation 4, we define $\bar{\alpha} \coloneqq \max_{k=1,\dots,N_T+1} \alpha_k$. Recall that $\bar{\Delta} = \max_{c\in\mathcal{C}, a\in\mathcal{A}, t\in[T]} (\max_{\pi\in\Pi} f_t(c, \pi(c)) - f_t(c, a))$. For any $k \in [N_T+1]$, if $(t - \tau_{k-1} - 1 \mod \lceil N_e/\alpha_k \rceil) \ge N_e$, then $A_t$ follows the stationary CB algorithm $\mathcal{B}$. Thus, the second term in equation 4 can then be decomposed as follows:

$$\mathbb{E}\left[\mathbb{1}\{\mathcal{G}_k\} \sum_{t=\nu_{k-1}}^{\nu_k - 1} \max_{\pi\in\Pi} f_t(C_t, \pi(C_t)) - f_t(C_t, A_t)\right]$$

$$\overset{(a)}{\le} \bar{\Delta}\ell_{k-1} + \bar{\Delta}N_e\left\lceil \frac{\nu_k - \nu_{k-1}}{\lceil N_e/\alpha_k \rceil}\right\rceil$$

$$+ \mathbb{E}\left[\mathbb{1}\{\mathcal{G}_k\} \sum_{t=\tau_{k-1}+1:(t-\tau_{k-1}-1) \mod \lceil N_e/\alpha_k \rceil \ge N_e}^{\nu_k - 1}\left(\max_{\pi\in\Pi} f_t(C_t, \pi(C_t)) - f_t(C_t, A_t)\right)\right]$$

$$\overset{(b)}{\le} \bar{\Delta}\ell_{k-1} + \bar{\Delta}\left[\alpha_k(\nu_k - \nu_{k-1}) + N_e\right] + R_{\mathcal{B}}(\nu_k - \nu_{k-1})$$

$$\le \bar{\Delta}\ell_{k-1} + \bar{\Delta}\left[\bar{\alpha}(\nu_k - \nu_{k-1}) + N_e\right] + R_{\mathcal{B}}(\nu_k - \nu_{k-1}) \tag{23}$$

where in step $(a)$, the first term bounds the regret due to the delay of the change detector, and the second term bounds the regret incurred due to forced exploration. In step $(b)$, as the reward samples in the history of the stationary bandit algorithm $\mathcal{B}$ are independent of those in $\cup_{(c,i)\in\mathcal{C}\times[N_e]}\mathcal{H}_{(c,i)}$, and that $\mathcal{G}_k$ only depends on samples in $\cup_{(c,i)\in\mathcal{C}\times[N_e]}\mathcal{H}_{(c,i)}$, the regret bound of the stationary bandit We also apply the fact that $R_{\mathcal{B}}(T)$ is increasing with $T$. Then, we can plug equation 23 and equation 22 into equation 4 and obtain:

$$R_T$$

$$\le \sum_{k=1}^{N_T+1} \bar{\Delta}(\nu_k - \nu_{k-1})\left(|\mathcal{C}|N_e k \delta_{\mathrm{F}} + (k-1)\left(2T^{-1} + \delta_{\mathrm{D}}\right)\right)$$

$$+ \sum_{k=1}^{N_T+1}\left(\bar{\Delta}\ell_{k-1} + \bar{\Delta}\left[\bar{\alpha}(\nu_k - \nu_{k-1}) + N_e\right] + R_{\mathcal{B}}(\nu_k - \nu_{k-1})\right)$$

$$\le \sum_{k=1}^{N_T+1} \bar{\Delta}(\nu_k - \nu_{k-1})\left(|\mathcal{C}|N_e(N_T+1)\delta_{\mathrm{F}} + N_T\left(2T^{-1} + \delta_{\mathrm{D}}\right)\right)$$

$$+ \sum_{k=1}^{N_T+1}\left(\bar{\Delta}\ell_{k-1} + \bar{\Delta}\left[\bar{\alpha}(\nu_k - \nu_{k-1}) + N_e\right] + R_{\mathcal{B}}(\nu_k - \nu_{k-1})\right)$$

$$= \bar{\Delta}T|\mathcal{C}|N_e(N_T+1)\delta_{\mathrm{F}} + 2\bar{\Delta}N_T + \bar{\Delta}TN_T\delta_{\mathrm{D}} + \bar{\Delta}\sum_{k=1}^{N_T}\ell_k + \bar{\Delta}\left[\bar{\alpha}T + (N_T+1)N_e\right]$$

$$+ \sum_{k=1}^{N_T+1} R_{\mathcal{B}}(\nu_k - \nu_{k-1})$$

$$\overset{(a)}{\le} \bar{\Delta}T|\mathcal{C}|N_e(N_T+1)\delta_{\mathrm{F}} + 2\bar{\Delta}N_T + \bar{\Delta}TN_T\delta_{\mathrm{D}} + \bar{\Delta}\sum_{k=1}^{N_T}\ell_k + \bar{\Delta}\left[\bar{\alpha}T + (N_T+1)N_e\right]$$

$$+ (N_T+1)R_{\mathcal{B}}\left(\frac{T}{N_T+1}\right). \tag{24}$$

In step $(a)$, we apply Jensen's inequality to the concave function $R_{\mathcal{B}}$. This concludes the proof of Theorem 4.8.

### B.6 Proof of Corollary 4.9

In PS-PBs, $N_e = d$, $p \geq 1/2$, and $q = r = 0$. Thus, $R_T = \tilde{\mathcal{O}}(\sqrt{dN_TT} + d^p\sqrt{N_TT}) = \tilde{\mathcal{O}}(d^p\gamma_T^q(\log|\Pi|)^r\sqrt{N_TT})$.

In PS-KBs, $q \geq 1/2$, $p \geq 0$ and $r = 0$. We can upper bound $N_e$ using the fact that $|\mathcal{V}_T| \leq \lceil\sqrt{d}R/2\delta_T\rceil^d$. Thus, $N_e \leq \lceil C\gamma_T^{2q/d}\rceil^d$ with $\delta_T = \frac{Rd^{1/2-2p/d}}{2(C\gamma_T^{2q})^{1/d}}$ and $R_T = \tilde{\mathcal{O}}((d^{2p}\gamma_T^{2q}N_TT)^{1/2} + d^p\gamma_T^q\sqrt{N_TT}) = \tilde{\mathcal{O}}(d^p\gamma_T^q(\log|\Pi|)^r\sqrt{N_TT})$.

We emphasize that when the number of action is smaller than the covering number, i.e., $|\mathcal{A}| < \lceil C\gamma_T^{2q/d}\rceil^d \leq \gamma_T$, then we can set $\mathcal{A}_e$ to be the entire action set $\mathcal{A}$. In this case, $N_e < \gamma_T$, guaranteeing order-optimal regret.

In PS-CBs, $N_e \leq |\mathcal{A}|$, $r \geq 1/2$, $p = q = 0$, and $|\Pi| = |\mathcal{A}|^{|\mathcal{C}|}$. Thus, $R_T = \tilde{\mathcal{O}}((|\mathcal{A}|\log|\Pi|)^r\sqrt{N_TT} + \sqrt{|\mathcal{C}||\mathcal{A}|N_TT}) = \tilde{\mathcal{O}}(d^p\gamma_T^q(|\mathcal{A}|\log|\Pi|)^r\sqrt{N_TT})$.

