# OpenReview forum: "DAL: A Practical Prior-Free Black-Box Framework for Non-Stationary Bandits"
_ICLR.cc/2026/Conference — Submitted to ICLR 2026_

### Official Review · Reviewer_TeBX · 2025-10-26

**Soundness:** 3
**Presentation:** 3
**Contribution:** 2
**Rating:** 4
**Confidence:** 3

**Summary:**

This paper proposes Detection Augmented Learning (DAL), a parameter-free, black-box framework for non-stationary bandits. DAL takes a stationary bandit algorithm and a change-point detection subroutine as inputs, and, through a forced-exploration mechanism, it adapts to non-stationary environments without requiring prior information. Extensive experiments on multiple benchmarks are conducted to validate the effectiveness of the proposed framework.

**Strengths:**

- This paper proposes a general framework for non-stationary bandits and establishes order-optimal regret guarantees in the piecewise-stationary setting. For the drifting case, the paper provides partial insights.
- The experimental evaluation is _thorough and diverse_, covering multiple bandit setups and including realistic datasets, which enhances the practical significance of the work.

**Weaknesses:**

- From a theoretical perspective, the main idea of augmenting a stationary bandit algorithm with a change-point detection module has been explored in prior work, limiting the conceptual novelty.

- Although the framework is claimed to extend naturally to contextual bandits, this case is not rigorously analyzed.

- The analysis for the drifting case remain limited, which constrains the overall contribution of the framework.

- Some assumptions, such as those in Proposition 4.2, require clearer justification or guidance on how they can be verified in practice.

**Questions:**

- How sensitive is DAL to the choice of covering set $A_e$ in large continuous action spaces?
- DAL depends critically on a GLR-type change detector, but the implementation specifics are not fully described, e.g., what is the exact testing statistics and threshold used for triggering restarts? How are the false alarms controlled?


I find the paper’s practical relevance to be stronger than its theoretical depth, and I would appreciate it if the authors could clarify the points raised above.

---

> ### Author Response · Authors · 2025-11-17
> **Response to Reviewer TeBX (Part 1)**
>
> We sincerely thank Reviewer TeBX for their thoughtful and constructive feedback.
>
> At a high level, our goal is to provide a practical, prior-free, black-box framework for non-stationary bandits that (i) attains strong empirical performance across multiple setups and (ii) comes with order-optimal regret guarantees. As the reviewer notes, the practical relevance of our work is strong; yet we  also focus on theoretical depth, developing novel analysis on the PS setting, and improving the state-of-the-art, while providing partial insights and empirical evidence in drifting environments.
>
> We address the main points raised in the review below.
>
> **Weaknesses**
>
> **Weakness 1**:  We agree that the idea of augmenting a stationary bandit algorithm with a change detector appears in prior work. Our contribution is to turn this idea into a *practical, prior-free, black-box framework* that is:
>
> - *Modular*: DAL only assumes a stationary bandit algorithm and a change detector that satisfy transparent conditions; the two components are analyzed and designed in a decoupled way.
>
> - *Prior-free and reward-based*: Unlike prior-free methods such as MASTER, ADA-OPKB, and ADA-ILTCB+, which detect changes via regret violations or optimal-policy switches (often impractical in realistic scenarios, as discussed in Gerogiannis et al., 2025), DAL detects changes directly through reward-distribution shifts.
> - *Also black-box in terms of detector*: DAL can be combined with any change detector satisfies a generic condition (Property 4.1), rather than relying on a specific GLR-type test. Our additional experiments with the GSR detector, which also satisfies Property 4.1, show comparable or slightly better performance than GLR. This detector-agnostic design is a key novelty over existing detection–restart methods, which typically hard-wire a specific detector (including MASTER), whereas DAL is black-box both in the choice of base bandit algorithm and in the choice of detector.
> - *General across settings*: The same DAL wrapper applies to parametric bandits, kernel bandits, and contextual bandits, with theoretical guarantees in the PS setting and strong empirical performance in drifting environments.
>
> While the overall “algorithm + detector” concept is classical, we believe this modular prior-free wrapper plus its PS analysis is substantively new.
>
> **Weakness 2**: We would like to clarify that our problem formulation in the introduction is essentially based on a contextual bandit formulation, and our analysis is built on that foundation. We agree that, in the original submission, the rigorous treatment of the contextual case in the proof of our main Theorem (Theorem 4.8 in the new draft) was not sufficiently visible.
>
> In the revised version, we have added a full, rigorous analysis and proof of our problem formulation (including the contextual case) in the appendix. Due to space limitations, we are unable to include the complete proof sketch in the main paper, but we believe this choice maintains a good balance between readability and rigor. We kindly encourage the reviewer to refer to the revised proof, and we would be happy to further expand this discussion if space permits.
>
> **Weakness 3**: As also discussed for other reviewers, we agree that our theoretical guarantees currently focus on the PS setting. The analysis of stationary bandit algorithms in general drifting environments is significantly more challenging, and existing tools do not directly extend to that regime. Our empirical results in drifting environments are therefore presented as strong *practical* evidence of DAL’s robustness, rather than as a proven optimal-theory result.

---

> > ### Author Response · Authors · 2025-11-17
> > **Response to Reviewer TeBX (Part 2)**
> >
> > **Weakness 4**: As mentioned in Section 3 of the original submission, limited space forced us to defer many practical implications of the propositions to the appendix. With the additional rebuttal page, we have now brought practical guidance into the main text (Section 3) and revised Proposition 4.3 (in the new draft) following Reviewer xZGQ’s suggestion.
> >
> > Concretely, we now explain how the results of the propositions are used in practice to construct the covering set $\\mathcal{A}\_e$:
> >
> > - *NS-PBs*: Following Proposition 4.2 (in the new draft), we greedily select linearly independent actions until we collect $d$ such vectors, or as many as exist if fewer are available.
> >
> > - *NS-KBs*: We construct a $\\delta\_T$-cover and choose $\\mathcal{A}\_e$ as the set of covering centers, as per Proposition 4.3 and Corollary 4.9 (in the new draft). In finite action spaces, we compute $\\gamma\_T$; if $|\\mathcal{A}| \\leq \\gamma\_T$, then the full action set already forms a valid cover and we take $\\mathcal{A}\_e = \\mathcal{A}$. Otherwise, we select the $\\gamma\_T$ actions closest to the cover centers. In all our NS-KB experiments, $\\gamma\_T$ is larger than $|\\mathcal{A}|$, so we always have $\\mathcal{A}\_e = \\mathcal{A}$.
> >
> > - *PS-CBs*: The action space is finite, and, as noted in Remark 4.4 (in the new draft), in our experiments we simply use $\\mathcal{A}\_e = \\mathcal{A}$.
> >
> > **Questions**
> >
> > **Question 1**: We thank the reviewer for this insightful question. As shown in Algorithm 1, DAL’s forced exploration depends on $N\_e$, the cardinality of $\\mathcal{A}\_e$. Intuitively, a larger $N\_e$ increases the exploration burden, since DAL must pull more actions to detect changes.
> >
> > - *NS-PBs*: In the NS-PB setting, Proposition 4.2 (in the new draft) shows that the cardinality of a suitable covering set is at most $d$. Thus, even if the underlying action space is infinite, DAL only needs to explore at most $d$ actions in $\\mathcal{A}\_e$. In this sense, DAL is not sensitive to the *size* of the continuous action space: it pays only a $d$-dependent cost. If there are multiple choices of $d$ linearly independent actions, the practical performance depends on the induced change magnitude $\\Delta\_c$ (as discussed in Section 4.1). For a fixed non-stationarity model, some choices of $d$ actions may yield larger $\\Delta\_c$, improving pre- and post-change sample complexity. However, our regret analysis accounts for the worst case over $\\Delta\_c$, so, at the theoretical level, DAL is not sensitive to which particular $d$ actions are chosen.
> >
> > - *NS-KBs*: In the NS-KB setting, the sensitivity of DAL is governed by the smoothness of the RKHS. If the Lipschitz constant $BL\_u$ is small, the RKHS contains smooth functions, so we can use a relatively large $\\delta\_T$, leading to a smaller covering set $\\mathcal{A}\_e$ (and thus a smaller $N\_e$). If the RKHS contains less smooth functions (larger $BL\_u$), we require a smaller $\\delta\_T$ to detect changes reliably, which increases $N\_e$. Nevertheless, to attain order-optimality DAL only needs to explore at most $\\gamma\_T$ actions, which is finite and significantly smaller than the (possibly infinite) original action space. As in NS-PBs, DAL is more sensitive to the underlying function class (smoothness) than to the raw size of the continuous action space.
> >
> > - *NS-CBs*: In the NS-CB/PS-CB case, the action set is finite, and one must fully explore all actions in order to characterize changes in the reward function, since the rewards can be completely uninformative about structural properties beyond their realized values.
> >
> > In all cases, DAL limits the cardinality of $\\mathcal{A}\_e$ to the minimum number of actions needed to characterize the reward function for detection and learning. These cardinalities match the quantities appearing in the minimax stationary regret bounds (e.g., $d$ for PBs, $\\gamma\_T$ for KBs, and $|\\mathcal{A}|$ for CBs). This principle guided our design of the covering-set selection procedures.

---

> > > ### Author Response · Authors · 2025-11-17
> > > **Response to Reviewer TeBX (Part 3)**
> > >
> > > **Question 2**: As mentioned in our response to Reviewer xZGQ and in the revised paper, space constraints in the original submission forced us to defer the implementation details and theory of the GLR detector to the appendix. With the rebuttal page, we have updated the paper so that Section 4.1 now:
> > >
> > > - states the theoretical requirements that any detector must satisfy to be compatible with DAL and achieve optimal regret, and
> > > - presents the implementation details of the GLR test (Algorithm 2) and an additional change detector, the Generalized Shiryaev–Roberts (GSR) test (Algorithm 3), for which we also provide new experiments to demonstrate DAL’s effectiveness.
> > >
> > > Due to space limits, the full general mathematical derivations of GLR and GSR remain in the appendix, but the main paper now includes their algorithmic forms and how they are used within DAL.
> > >
> > > The false alarms are controlled by the false-alarm probability $\\delta\_{\\mathrm{F}}$, which is a design parameter of the detector. To ensure good performance, the detector $\\mathcal{D}$ must (i) minimize the expected detection delay after a change, while (ii) keeping the false-alarm probability small. Property 4.1 (in the new draft) formalizes the relationship between $\\delta\_{\\mathrm{F}}$ and the detector’s performance. In practice, we set $\\delta\_{\\mathrm{F}}$ as an input to the detector, and Theorem 4.8 (in the new draft) shows that it suffices to choose $\\delta\_{\\mathrm{F}} = T^{-\\gamma}$ for some $\\gamma > 1$ to obtain order-optimal regret in our setting.
> > >
> > > For completeness, we summarize the GLR and GSR statistics below. Let $\\{X\_1, \\dots, X\_n\\}$ be the sequence of observed rewards and let $f\_\\theta$ denote the density of a Gaussian random variable with mean $\\theta \\sigma^{2}$ and variance $\\sigma^{2}$ or a Bernoulli random variable with the same mean. The GLR statistic is
> > > $$
> > > G\_n \\coloneqq \\sup\_{s \\in [n]} \\log\\left( \\frac{ \\sup\_{\\theta\_0,\\theta\_1 \\in \\mathbb{R}} \\prod\_{i=1}^{s} f\_{\\theta\_0}(X\_i) \\prod\_{i=s+1}^{n} f\_{\\theta\_1}(X\_i) }{ \\sup\_{\\theta \\in \\mathbb{R}} \\prod\_{i=1}^{n} f\_{\\theta}(X\_i) } \\right),
> > > $$
> > > and the GSR statistic is
> > > $$
> > > W\_n \\coloneqq \\frac{1}{n} \\sum\_{s=1}^{n} \\left( \\frac{ \\sup\_{\\theta\_0,\\theta\_1 \\in \\mathbb{R}} \\prod\_{i=1}^{s} f\_{\\theta\_0}(X\_i) \\prod\_{i=s+1}^{n} f\_{\\theta\_1}(X\_i) }{ \\sup\_{\\theta \\in \\mathbb{R}} \\prod\_{i=1}^{n} f\_{\\theta}(X\_i) } \\right).
> > > $$
> > >
> > > A GLR-based restart is triggered at the smallest $n$ such that
> > > $$
> > > G\_n \\geq \\beta(n,\\delta\_{\\mathrm{F}}),
> > > $$
> > > and similarly a GSR-based restart is triggered at the smallest $n$ such that
> > > $$
> > > W\_n \\geq n \\cdot \\exp\\bigl(\\beta(n,\\delta\_{\\mathrm{F}})\\bigr).
> > > $$
> > > The threshold function $\\beta(n,\\delta\_{\\mathrm{F}})$ depends on the number of samples $n$ and the false-alarm probability $\\delta\_{\\mathrm{F}}$. A typical choice, following Huang et al. (2025) and Besson et al. (2022), is
> > > $$
> > > \\beta(n,\\delta\_{\\mathrm{F}}) = 6\\log\\bigl(1 + \\log n\\bigr) + \\frac{5}{2} \\log\\left( \\frac{4 n^{3/2}}{\\delta\_{\\mathrm{F}}} \\right) + 11.
> > > $$
> > > Thus, smaller $\\delta\_{\\mathrm{F}}$ leads to larger thresholds and, in practice, fewer false alarms but potentially longer detection delays.
> > >
> > > ---
> > >
> > > **Finally, we kindly refer to our General Response comment where we list all the revisions done in our draft.**
> > >
> > > We hope to have addressed all of the reviewer’s questions to the best of our ability. Nevertheless, we would be very happy to receive further feedback if there are additional concerns. We hope the reviewer will kindly consider our clarifications and the contributions of our work.

---

> ### Author Response · Authors · 2025-11-26
> **Kind Reminder**
>
> Dear Reviewer TeBX,
>
> Thank you again for your thoughtful review and your feedback on our submission. Since the discussion period is progressing, we wanted to follow up to ensure that there is sufficient time to engage with any remaining questions and to confirm that our rebuttal has fully addressed your concerns. We would greatly appreciate it if you could share your thoughts or highlight any remaining issues you feel require additional clarification.
>
> Thank you once again for your time and consideration.

---

### Official Review · Reviewer_V4aw · 2025-10-27

**Soundness:** 3
**Presentation:** 3
**Contribution:** 2
**Rating:** 4
**Confidence:** 3

**Summary:**

For non-stationary bandits, most existing methods — such as restart, weighted/discounted, and sliding-window methods — can get good empirical performance and near-optimal regret guarantees. However, they rely on strong prior knowledge about the non-stationarity of environment. In contrast, MASTER achieves optimal regret without requiring such prior knowledge, but it is very complex: it runs many learners in parallel, which makes it hard to use in practice and often weak in experiments.

This paper focuses on the piecewise-stationary setting and proposes a black-box method that achieves (near-)optimal regret and strong empirical performance. The method keeps a small covering set of arms, occasionally pulls arms from this set to detect changes, and restarts the base learner when a change is detected. This removes the need to know the degree of non-stationarity and avoids maintaining many parallel learners.

**Strengths:**

1. The method provides an algorithm with theoretical guarantees that does not rely on prior knowledge of the environment, and it also shows strong empirical performance.
2. The method is general: it acts as a black-box change detector that can be wrapped around different types of bandit algorithms, and it works across multiple bandit settings.

**Weaknesses:**

1. The method does not provide theoretical guarantee for the drifting case. This is expected, because the change-detection mechanism is designed for abrupt changes, not for drifting changes. The paper only shows empirical performance on drifting, but bandits are primarily a theoretical setting, so having a matching optimal regret guarantee there is important and is currently missing.

2. Compared to MASTER, this paper’s analysis in the piecewise-stationary setting relies on an extra assumption:  changes in the environment must be separated by a sufficiently long stable period. This assumption appears inside Theorem 4.4, but it is not stated clearly as its own assumption. I suggest the authors make this assumption explicit and discuss it up front. Otherwise, the comparison to prior work (MASTER) is not fair, and the assumption feels too hidden.

**Questions:**

1. The paper repeatedly uses the broad term “non-stationary bandits,” but after reading the paper, the theory really only covers the piecewise-stationary case. For drifting, there is no matching theoretical analysis, but only experiments. By this standard, any prior piecewise-stationary bandit method could also run on a drifting simulation and then claim to solve “non-stationary bandits,” which would be an overclaim. Since the proposed method is not specifically designed for drifting, I believe the paper (including the title) should make it explicit that the setting is piecewise-stationary, not general non-stationary.

2. Prior work on piecewise-stationary bandits already has prior-free detection-and-restart methods. It is not yet clear to me what the real difficulty is in turning those approaches into a black-box wrapper, and how this paper goes beyond that in a substantive way.

I would be happy to raise my score if the authors can make the requested revisions and clarify these points.

---

> ### Author Response · Authors · 2025-11-17
> **Response to Reviewer V4aw (Part 1)**
>
> We sincerely thank Reviewer V4aw for their thoughtful and constructive feedback.
>
> Below we address the main points raised in the review.
>
> **Questions**
>
> **Question 1**: We fully agree with the reviewer’s point, and in response, we have revised the paper to explicitly describe our approach as tailored for PS. The general term “non-stationary bandits” is now reserved for high-level motivation and for discussing the broader context in which both PS and drifting environments arise. Wherever appropriate, including in the title, abstract and the paper, we now explicitly reflect that our setting corresponds to the PS setting.
>
> Our original goal was to introduce a practical, prior-free method for non-stationary bandits by combining a strong change detector with a bandit algorithm, inspired by the open problem of Gerogiannis et al. (2025). After observing strong empirical performance of DAL in both PS and drifting environments, we sought to explain both types of non-stationarity theoretically. Because analyzing stationary bandit algorithms under drift is difficult, our current theoretical guarantees are restricted to the PS setting.
>
> **Question 2**: As also mentioned in our response to Reviewer xZGQ, our goal is to advance non-stationary bandit algorithm design by addressing two main gaps: the lack of prior-free practical methods with strong empirical performance, and the absence of a black-box that achieves this. In the prior-free detection-and-restart literature, achieving order-optimal regret in the settings we consider is, to the best of our knowledge, limited essentially to MASTER, ADA-OPKB, and ADA-ILTCB+.
>
> MASTER, ADA-OPKB, and ADA-ILTCB+ detect non-stationarity via regret-bound violations or optimal-policy switches, mechanisms that can be non-practical in realistic environments (Gerogiannis et al., 2025). Our work, to the best of our knowledge, is the first prior-free black-box wrapper that departs from this paradigm and instead detects non-stationarity directly through changes in the reward distributions.
>
> Outside these prior-free, state-of-the-art detection-based methods, existing approaches are generally not directly extensible to a black-box wrapper: their detectors are tailored to specific algorithms and settings, and the analysis couples the detector tightly to that structure. Hence, it is unclear how to generalize such detectors into a modular wrapper. Even for methods that use standard stationary bandit algorithms as components, the detection mechanism is often specific to the chosen algorithm. DAL can be combined with any change detector that satisfies a generic condition (Property 4.1 in the revised draft), rather than relying on a specific GLR-type test. To illustrate this, we add experiments with an additional detector, the GSR test, which also satisfies Property 4.1 and achieves performance comparable to or better than GLR. This flexibility is a key novelty compared to existing detection–restart methods. DAL is black-box not only with respect to the stationary bandit algorithm and across different bandit settings, but also with respect to the choice of detector.
>
> A key contribution of our work is to decouple the roles of the bandit algorithm and the detector so they operate in a modular, independent fashion. We theoretically analyze how regret decomposes across the interactions between forced exploration, covering sets, the change detector, and the input bandit algorithm—both for finite and continuous action spaces. While tools exist for analyzing each component individually, combining them into a general prior-free black-box framework is non-trivial and central to our analysis. Finally, the DAL paradigm is a modular structure that naturally extends to broader online learning settings with similar dynamics, laying the foundation for general detection-augmented learning strategies.

---

> ### Author Response · Authors · 2025-11-17
> **Response to Reviewer V4aw (Part 2)**
>
> **Weaknesses**
>
> **Weakness 1**: We agree that the bandit literature often emphasizes theoretical analysis over practical performance. A main goal of our work is to empirically validate order-optimal approaches and push bandit algorithms beyond theory through diverse real-world experiments. We concede that the missing theoretical analysis is important, but we argue that methods must also have strong practical performance, not exist only for theory, a core message of our paper. Our empirical results on drifting environments suggest that DAL can be effective beyond PS, but without a matching regret analysis this is not a solved theoretical case.
>
> An additional insight is that the challenge in the drifting case does not come from the change detector, which can be adapted to detect shifts in drifting non-stationarity. The bottleneck is analyzing the stationary algorithm, which essentially requires the non-stationarity to approach zero as time goes to infinity, an assumption in both MASTER and WeightUCB.
>
> **Weakness 2**: We thank the reviewer for noting that the separation assumption was not sufficiently highlighted. In the original submission, space constraints led us to state it inside our main Theorem and then discuss it in Remark 4.6 (in the old draft) and the following paragraph. In the revised draft, we therefore introduce the separation assumption as its own standalone assumption, alongside a supplementary definition before the main Theorem (Theorem 4.8 in the new draft). We then add an additional paragraph to explain the assumption in more depth and keep the accompanying explanatory paragraph and Remark directly below this assumption, making its role and implications explicit.
>
> ---
>
> **Finally, we kindly refer to our General Response comment where we list all the revisions done in our draft.**
>
> We hope to have addressed all of the reviewer’s questions to the best of our ability. Nevertheless, we would be very happy to receive further feedback if there are additional concerns. We hope the reviewer will kindly consider our clarifications and the contributions of our work.

---

> ### Author Response · Authors · 2025-11-26
> **Kind Reminder**
>
> Dear Reviewer V4aw,
>
> Thank you again for your thoughtful review and your feedback on our submission. Since the discussion period is progressing, we wanted to follow up to ensure that there is sufficient time to engage with any remaining questions and to confirm that our rebuttal has fully addressed your concerns. We would greatly appreciate it if you could share your thoughts or highlight any remaining issues you feel require additional clarification.
>
> Thank you once again for your time and consideration.

---

### Official Review · Reviewer_fy93 · 2025-11-04

**Soundness:** 3
**Presentation:** 2
**Contribution:** 2
**Rating:** 4
**Confidence:** 3

**Summary:**

This paper focuses on a classical problem, that of learning in non-stationary bandits. The idea, essentially, is to augment a standard bandit algorithm with a "change detector". Classical bandit algorithms have their theory (and presumed applications) made under the stationarity assumption, which is not necessarily true in practice. Such changes can take the form of both abrupt and gradual changes. The authors propose a framework based on (1) detecting change of distribution by considering shifts in mean action rewards (2) forced exploration according to a schedule, forcing the bandit algorithm to essentially "drift" in state space. The mean-action shift is done by choosing an "appreciable" mean shift, exploiting some structure of the problem in deciding on which one. Some theoretical results on regret are provided.

**Strengths:**

Strengths: this is a nice problem, and one that has been considered by many authors over the years. The approach, while fairly simple, is effective. The experiments seem to be justifiable and demonstrate the performance of the method.

**Weaknesses:**

Weaknesses: the paper is not so easy to digest and understand at times. The tuning of the methods seems challenging, and the authors do not convince the reader otherwise. No details on the construction of the covering set are provided, as an instance.

Questions: what if the process contains a mix of abrupt and gradual changes? Can this method be augmented with memory, allowing to go back to previous regimes, instead of effectively starting from scratch every time?

**Questions:**

N/A

---

> ### Author Response · Authors · 2025-11-17
> **Response to Reviewer fy93 (Part 1)**
>
> We sincerely thank Reviewer fy93 for their thoughtful and constructive feedback.
>
> **Revisions and Clarifications**: During the rebuttal, we were able to test DAL with an additional change detector, the Generalized Shiryaev–Roberts (GSR) test, and we have included these new results in our revised draft. Importantly, to enhance the clarity of our work, we also revised several sections by adding further details on both our theory and experiments. In particular, we highlighted the theoretical conditions required for DAL’s change detector, added the algorithmic implementations of the GLR and GSR tests in Section 4.1, and isolated the conditions associated with our main Theorem (Theorem 4.8 in the new draft).
>
> Below we address the main points raised in the review.
>
> **Weaknesses**
>
> **Tuning**: As mentioned in our paper, the state-of-the-art methods we compare DAL with are tuned according to their original works and any publicly available code.  Due to space constraints in the original submission, implementation and tuning details had to be deferred to the appendix, as mentioned in Section 3. With the additional page provided for the rebuttal, we have now moved all necessary details into Section 3.
>
> First, the stationary algorithm used as input to DAL is tuned exactly as proposed in the original works. Second, we clarify how the covering set is constructed in practice for each setting:
>
> - *NS-PBs*: Following Proposition 4.2 (in the new draft), we greedily select linearly independent actions until we collect $d$ such vectors, or as many as exist if fewer are available.
>
> - *NS-KBs*: The covering set is obtained from a $\\delta\_T$-cover by choosing the centers of the covering balls, as per Proposition 4.3 and Corollary 4.9 (in the new draft). In finite action spaces, we compute $\\gamma\_T$; if $|\\mathcal{A}| \\leq \\gamma\_T$, then by Corollary 4.8 the full action set already forms a valid cover and we take $\\mathcal{A}\_e = \\mathcal{A}$. Otherwise, we select the $\\gamma\_T$ actions closest to the cover centers. In all our NS-KB experiments, $\\gamma\_T$ is larger than $|\\mathcal{A}|$, so we always have $\\mathcal{A}\_e = \\mathcal{A}$.
>
> - *PS-CBs*: Here the action space is finite, and, as noted in Remark 4.4 (in the new draft), in all our experiments we simply take $\\mathcal{A}\_e = \\mathcal{A}$.
>
> Finally, we moved the algorithmic implementations of both the GLR and GSR tests into the main paper and made explicit our choice of thresholds:
> - For GLR, we set: $ \\beta\_{\\mathrm{GLR}}(n,\\delta\_{\\mathrm{F}}) = \\log\\bigl(n^{3/2}/\\delta\_{\\mathrm{F}}\\bigr)$,
> - For GSR, we set $\\beta\_{\\mathrm{GSR}}(n,\\delta\_{\\mathrm{F}}) = n^{5/2}/\\delta\_{\\mathrm{F}}$,
>
> with $\\delta\_{\\mathrm{F}} = 1/\\sqrt{T}$, as per Huang et al. (2025) and Besson et al. (2022).

---

> > ### Author Response · Authors · 2025-11-17
> > **Response to Reviewer fy93 (Part 2)**
> >
> > **Questions**
> >
> > **Question 1**: We thank the reviewer for this interesting question. If the process contains a mix of abrupt and gradual changes, it can be viewed as a special case of a gradually changing process, in the sense that there is a non-zero change at each time-step. In such a setting, our hypothesis for drift environments could potentially apply, but the validity of our guarantees depends heavily on how the gradual component behaves.
> >
> > For example, if the gradual changes occur at a very fast rate, then our hypothesis is likely to fail due to the excessive level of non-stationarity at each time-step, which would prevent the absorption into the noise. More generally, this mixed scenario raises several non-trivial questions (e.g., what is the admissible range of per-step changes that still allows change detection and learning to be effective, and what type of detector is best suited to this regime). We agree that this is a very intriguing setting, but it would require substantial new analysis beyond the scope of the current work.
> >
> > **Question 2:** We also appreciate the suggestion of augmenting DAL with memory, allowing the algorithm to return to previously encountered regimes instead of restarting from scratch. Conceptually, one could combine a change detector to signal a regime shift with a mechanism that attempts to match the new regime to one stored in memory. However, designing such a mechanism in a regret-efficient way is highly non-trivial.
> >
> > In particular, selecting the appropriate regime from memory corresponds to a composite hypothesis testing problem, where the composite test (over many possible past regimes) is not known in advance. This raises challenging questions about how to balance exploration, exploitation, and regime identification, and how to do so while preserving strong regret guarantees. We believe that incorporating memory into non-stationary bandit algorithms is a very promising research direction, but it falls outside the scope of the current work.
> >
> > ---
> >
> > **Finally, we kindly refer to our General Response comment where we list all the revisions done in our draft.**
> >
> > We hope to have addressed all of the reviewer’s questions to the best of our ability. Nevertheless, we would be very happy to receive further feedback if there are additional concerns. We hope the reviewer will kindly consider our clarifications and the contributions of our work.

---

> ### Author Response · Authors · 2025-11-26
> **Kind Reminder**
>
> Dear Reviewer fy93,
>
> Thank you again for your thoughtful review and your feedback on our submission. Since the discussion period is progressing, we wanted to follow up to ensure that there is sufficient time to engage with any remaining questions and to confirm that our rebuttal has fully addressed your concerns. We would greatly appreciate it if you could share your thoughts or highlight any remaining issues you feel require additional clarification.
>
> Thank you once again for your time and consideration.

---

### Official Review · Reviewer_xZGQ · 2025-11-12

**Soundness:** 2
**Presentation:** 2
**Contribution:** 2
**Rating:** 4
**Confidence:** 4

**Summary:**

This work focus on the regret minimization problem in non-stationary bandits. It proposed the DAL technique to detect unknown changes in the environment. Both numerical experiments and theoretical analysis are presented in this work.

**Strengths:**

1. Many related works are discussed.
2. Numerical experiments are done in various datasets.

**Weaknesses:**

This work presents a set of numerical results and a set of analytical results while neither of them fully convince me the superiority of the algorithm. I wonder what is the key contribution/focus of the work. Some key concerns are as below:
1. Abstract: It is claimed that 'DAL accepts any stationary bandit algorithm as input' while Propositions/theorems (e.g. Theorem 4.4) come with some assumptions/conditions. It is somehow confusing.
1. Line 28: It is claimed that 'MABs fall into ... PB, NPB, CB'. I feel maybe it is not that proper to say so. For example, contextual bandits can also be viewed as a parametric setting from some perspective.
1. Algorithm 1: I think the algorithm is a key contribution of this work, while the pseudocode is not that easy to understand.
  1. What is $N_e$?
  1. When will $D( \ldots )=\text{detection}$ (in line 6)?
1. Many subplots in Figures 1 and 2 present the regret/reward of only a portion of discussed algorithms? Do those missing algorithms perform better than DAL? An explanation is appreciated.
1. Proposition 4.2: It is a bit unusal that the Lipschitz constant $BL_u$ does not affect the bound on $|V_T|$. Some explanations are appreciated.
1. Theorem 4.4 comes many conditions/assumptions without discussions. Besides, how the regrets stated in the paragraph beginning from Line 414 is not that clear. Some explanations here are also appreciated.


Besides, here is one minor suggestion:
1. The algorithms should be arranged in the same order in the legend box for Figures 1 and 2.

**Questions:**

See *Weaknesses* above.

---

> ### Author Response · Authors · 2025-11-17
> **Response to Reviewer xZGQ (Part 1)**
>
> We sincerely thank Reviewer xZGQ for their thoughtful and constructive feedback.
>
> **Minor Suggestion**
>
> We would like to thank the reviewer for their suggestion. We made sure the algorithms appear in the same order in both figures.
>
> **Clarifications on the Contributions**
>
> As stated in the introduction, we aim to advance non-stationary bandit algorithm design by addressing two gaps: the lack of prior-free practical methods with strong empirical performance and the absence of a black-box paradigm that systematically achieves this, given the limitations of MASTER.
>
> First, we highlight the superiority of DAL in practice and theory. Empirically, DAL is versatile across many non-stationary environments and bandit models and outperforms all state-of-the-art methods (not only MASTER) in extensive synthetic simulations and standard real-world benchmarks. Theoretically, DAL achieves order-optimal regret in PS settings, improving the bounds for PS-SCBs and PS-KBs and attaining the best bounds for the remaining settings.
>
> Our primary contribution is thus a practical prior-free black-box detection-based framework with provable guarantees. The first key novelty is to show how forced exploration, change detection, and any stationary bandit algorithm can be composed into a single black-box procedure. While each component is known individually, the open questions of how much to explore, which actions to probe, when to detect changes, and when to reset are addressed in a principled way. The second novelty is the DAL paradigm itself: a modular structure that extends to broader online learning settings where similar dynamics arise, laying the foundation for general detection-augmented learning strategies.
>
> Finally, unlike prior-free black-box or detection-restarting approaches for general bandits, DAL is the first to detect non-stationarity directly through changes in the reward distributions. This contrasts with the existing best methods, MASTER, ADA-OPKB, and ADA-ILTCB+, which rely on regret-bound violations or detecting optimal-policy switches, which can be non-practical in realistic environments (Gerogiannis et al., (2025)).
>
> Importantly, DAL can be combined with any change detector that satisfies a generic condition (Property 4.1 in the revised draft), rather than relying on a specific GLR-type test. To illustrate this, we add experiments with an additional detector, the GSR test, which also satisfies Property 4.1 and achieves performance comparable to or better than GLR. This flexibility is a key novelty compared to existing detection–restart methods: non–black-box approaches typically hard-wire a detector to a particular setting or base algorithm, while the only other existing black-box method, MASTER, is coupled to a specific detector. In contrast, DAL is black-box not only with respect to the stationary bandit algorithm and across different bandit settings, but also with respect to the choice of detector.

---

> > ### Author Response · Authors · 2025-11-17
> > **Response to Reviewer xZGQ (Part 2)**
> >
> > We address the main points raised by the Reviewer below.
> >
> > **Weakness 1**: We clarify that DAL can indeed have *any* stationary bandit algorithm as its input; though, order-optimal performance can't be guaranteed if the chosen algorithm is suboptimal. Typically, stationary bandit algorithms are tied to good performance (i.e., order-optimal regret). Since some stationary algorithms may not achieve optimal regret, Theorem 4.8 (in the new draft) makes this explicit by specifying the condition on its input's regret under which DAL attains order-optimal regret in the PS setting. However, if a stationary bandit algorithm has strong empirical performance but lacks a complete analysis, it can still be combined with DAL, without theoretical guarantees. We revised Section 4.2 to make this clearer.
> >
> > Finally, all propositions concerning the construction of $\mathcal{A}_e$ are independent of the bandit algorithm and depend only on the problem structure. These results describe solely how the covering set is formed.
> >
> > **Weakness 2**: We clarify that in Lines 25–28 we state that "many *variants of MABs* … fall into". Our intention was not to refer to classical MABs themselves, but specifically to their *variants*, which include the settings considered in our work. To avoid ambiguity, we revised the statement to make this explicit.
> >
> > We agree with the reviewer that CBs can often be viewed as parametric settings (e.g., linear CBs). Thus, our problem formulation is deliberately general and covers this case as well. Since state-of-the-art algorithms for CBs are applicable to any parametric contextual setting, we did not specify a particular CB variant.
> >
> > **Weakness 3**: We agree that the algorithm is a core contribution of our work, which is why Section 2 is dedicated to a detailed explanation. Algorithm 1 contains a single if–else structure: the first branch handles forced exploration and change detection, while the second branch runs the stationary bandit algorithm.
> >
> > As shown in Section 2, forced exploration is implemented via round-robin selection over the covering set using the exploration frequency $\alpha_k$. Whenever an action from $\mathcal{A}_e$ is selected, its reward is appended to the corresponding history, and the detection test is run. If a change is detected, we reset everything, including the histories and the stationary bandit algorithm. We revised the pseudocode and would greatly appreciate any further suggestions for clarity.
> >
> > **Weakness 4**: As defined in Section 2, $N_e$ denotes the size of $\mathcal{A}_e$, i.e., the number of actions used for forced exploration and for detecting changes. We made this more explicit in Algorithm 1.
> >
> > **Weakness 5**: As mentioned in Section 3, due to space constraints we deferred detector details to the appendix. With the additional rebuttal page, we now include a section dedicated to the detector, highlighting both its theoretical role in our analysis and its practical implementation (new Algorithm 2).
> >
> > To show that our framework’s performance is due to the *design* of DAL rather than a specific detector, we added new results for an alternative detector satisfying our theoretical requirements: the Generalized Shiryaev–Roberts (GSR) test, whose implementation is presented in the new Algorithm 3.
> >
> > Concretely, for the GLR test, given a history of reward samples $\{X\_1, \\dots, X\_n\}$, a detection is triggered if there exists $k \\in [n]$ such that
> > $$
> > \\mathrm{GLR}\_k := k\\,\\mathrm{kl}(\\hat{\\mu}\_{1:k}, \\hat{\\mu}\_{1:n}) + (n-k)\\,\\mathrm{kl}(\\hat{\\mu}\_{k+1:n}, \\hat{\\mu}\_{1:n}) \\geq \\beta\_{\\mathrm{GLR}}(n,\\delta\_{\\mathrm{F}}).
> > $$
> > $\\hat{\\mu}\_{t\_1:t\_2}$ denotes the empirical mean of the rewards from $t\_1$ to $t\_2$, $\\delta\_{\\mathrm{F}}$ is the false-alarm probability of the detector, and $\\beta\_{\\mathrm{GLR}}$ is the test threshold. Following Huang et al., 2025 and Besson et al., 2022, we can set $\\beta\_{\\mathrm{GLR}}(n,\\delta\_{\\mathrm{F}}) = \\log(n^{3/2})/\\delta\_\\mathrm{F}.$
> >
> > For the GSR test, a detection is triggered if
> > $$
> > \\sum\_{\\tau=1}^k \\exp\\bigl(\\mathrm{GLR}\_\\tau\\bigr) \\geq \\beta\_{\\mathrm{GSR}}(n,\\delta\_{\\mathrm{F}}) \\quad \\text{for some } k \\in [n],
> > $$
> > where we can choose $\\beta\_{\\mathrm{GSR}}(n,\\delta\_{\\mathrm{F}}) = n^{5/2}/\\delta\_{\\mathrm{F}}$, following Huang et al., 2025.
> >
> > We revised all detector-related details in Sections 3.2 and 4.1, moving the essential information from the appendix into the main text.

---

> > > ### Author Response · Authors · 2025-11-17
> > > **Response to Reviewer xZGQ (Part 3)**
> > >
> > > **Weakness 6**: As shown in Section 3, the algorithms we include in each experiment are the state-of-the-art methods for the settings. Among these, the only truly fair prior-free baselines are MASTER, ADA-OPKB, and ADA-ILTCB+, as they are designed specifically for prior-free non-stationary bandits. To the best of our knowledge, these are the only existing prior-free solutions for general non-stationary bandits, aside from our method. The other algorithms mentioned in the introduction are consistently outperformed by these state-of-the-art approaches in their respective regimes. We note that WeightUCB is the best *prior-based* method in the settings we consider. While our main focus is on prior-free benchmarks, we still include WeightUCB to benchmark DAL against the best prior-based approach. In all cases, we do not omit algorithms because they outperform DAL; we first confirmed that WeightUCB outperforms all alternatives, and since DAL outperforms WeightUCB, it also outperforms the omitted methods. We select the best baselines for clarity and readability of the plots.
> > >
> > > **Weakness 7**: We thank the reviewer for this point. Implicitly, the Lipschitz constant $BL_u$ does affect $\delta_T$, and thus the bound on $|V_T|$. Specifically, when $BL_u$ is small, functions in the RKHS are very smooth, so a larger $\delta_T$ can be used: the lower bound in Proposition 4.3 (in the new draft) becomes smaller, fewer points are needed to detect a change, and hence the cardinality of $V_T$ decreases.
> > >
> > > Thus, the cardinality of $V_T$ and $BL_u$ are linked through $\delta_T$, even though this dependence is not explicit in the original statement. To make this connection clear, we revised the proposition.
> > >
> > > **Weakness 8**: We clarify that our main Theorem involves two conditions: one on the separation of change-points and one on the regret of the input stationary algorithm. Due to space constraints, we stated both assumptions inside the theorem. The condition on the separation of the change-points is discussed in detail in Remark and in the paragraph immediately following it. As noted in our response to Weakness 1, the condition on the regret of the input algorithm is standard: to achieve order-optimal regret in the non-stationary setting, the underlying stationary algorithm must itself have order-optimal regret.
> > >
> > > Following the suggestion of Reviewer V4aw, we have now (i) presented the assumption on the separation of change-points as a standalone assumption, combined with a supplementary definition and followed an additional paragraph, the previous explanatory paragraph and the remark, and (ii) separately discussed the requirement on the regret of the input algorithm. We list the order-optimal regret bounds in the stationary settings we consider and clarify that, since DAL can accept any stationary algorithm, we require its regret bound $R_B(T)$ in Theorem 4.8 (in the new draft) to be concave and increasing in $T$.
> > >
> > > Regarding the reviewer’s statement “Besides, how the regrets...clear,” we are somewhat unsure which aspect is unclear and would greatly appreciate further clarification. In that paragraph, we summarize the state-of-the-art regret bounds in the PS setting and explain how DAL either improves or matches them; this discussion is intended as a direct follow-up to the opening statement of Section 4.2. In the revised draft, we have added an additional paragraph above Theorem 4.8 (in the new draft) to make this connection more explicit and to more clearly link the informal discussion to the formal theorem. We hope this clarifies the intent of the paragraph, and we would be very grateful for any further guidance on what remains unclear so that we can address it fully.
> > >
> > > ---
> > >
> > > **Finally, we kindly refer to our General Response comment where we list all the revisions done in our draft.**
> > >
> > > We hope to have addressed the reviewer’s questions fully. We would be more than happy to receive further feedback if there are additional concerns. We hope the reviewer will kindly consider our clarifications and the contributions of our work.

---

> ### Comment · Reviewer_xZGQ · 2025-11-23
>
> Thanks for your detailed response. My concerns have been partially solved and some further comments are as below:
> 1. Abstract: the current version still claim that 'DAL accepts any stationary bandit algorithm as input' while Propositions/theorems (e.g. Theorem 4.4) come with some assumptions/conditions.
>    1. Except for revising Section 4.2, I think the abstract should also be revised.
> 1. Thanks for your response.
> 1. Thanks for your response. However, besides more details in the appendix, you are still suggested to place some brief discussion of Line 6 ans mention that 'more details are in Appendix ***' in Section 2.
> 1. Why not compare all algorithms meanwhile? That should bring no harm but better clarity.
> 1. Proposition 4.2: thanks for the revision. It is clearer now.
> 1. Theorem 4.4: I think the discussions are important. You make try to find some place for it in the main part of the work.
>
> Besides, since you would like to change the title of the work, I wonder if another round is review is required and would like to keep the rating for the moment.

---

> ### Author Response · Authors · 2025-11-25
> **Addressing Remaining Concerns**
>
> We sincerely thank Reviewer xZGQ for their follow-up response and for engaging in a detailed discussion with us. Since the reviewer is currently keeping their score, we would like to further clarify the remaining concerns in the hope that our revisions address them more fully.
>
> **We have updated our draft according to the reviewer's concerns.**
>
> **Change of title.**
>
> As per the ICLR Author Guide, authors may change the title, abstract, and content as long as the changes are communicated to the reviewers and area chair. We have followed this guideline and explicitly described the change in our rebuttal and revised draft. Our original title used “non-stationary bandits” in line with prior work that also employs broad terminology for more specific settings (e.g., “A Simple Approach for Non-stationary Linear Bandits” by Zhao et al. 2020).
>
> To address **Reviewer V4aw’s concern** about potential overclaiming, we changed only one word, from “Non-Stationary” to “Piecewise-Stationary,” to more precisely reflect the theoretical scope. If the reviewer feels that the original broader wording was already acceptable, we are happy to follow whatever is deemed most appropriate.
>
> **1. Abstract statement.**
>
> In our response to Weakness 1, we explained in what sense DAL can accept *any* stationary bandit algorithm as input. The abstract is meant to convey this modularity at a high level, while the exact conditions appear in the theory. Theorem 4.4 (4.8 in the new draft) requires that the stationary algorithm has regret that may depend arbitrarily on problem-dependent constants but grows at most on the order of $\\sqrt{T}$ in the horizon. This covers all commonly used stationary bandit algorithms; algorithms with worse-than-$\\sqrt{T}$ regret are typically considered suboptimal and rarely used. At the same time, DAL can *practically* wrap algorithms that lack complete theory (e.g., UCB1-Tuned by Auer et al. 2002), but for such algorithms we do not claim theoretical guarantees.
>
> To align the abstract more closely with the theorem and with your suggestion, we have revised it to state explicitly that DAL “accepts any stationary bandit algorithm with order-optimal regret,”.
>
> **3. Brief discussion for Line 6.**
>
> Per your suggestion, we have added a short explanation in Section 2 and **explicitly direct the reader to Sections 3, 4.1 and to the appendix for the formal, rigorous definition**.
>
> **4. “Why not compare all algorithms?”**
>
> We fully agree that including more baselines can improve clarity, but there is also a trade-off with readability when too many curves appear in a single plot. Our original choice was mainly for presentation: plotting every algorithm from the related work section would make the figures crowded and harder to interpret.
>
> However, we have now expanded the synthetic experiments to **include all the additional related algorithms** that were previously only discussed, adding **11 additional algorithms**. To keep the figures interpretable, we group the same paradigms (discounted, budget-restart, and sliding-window families) while still giving separate entries to approaches that are meaningfully different. For the real-world experiments, we focus on the current state-of-the-art methods, since the additional algorithms are not competitive with these stronger baselines.
>
> If the reviewer feels it would still be useful, we are happy to extend the plots in the real-world data. Thus, **we also updated our code**.
>
> We would like to emphasize that the state-of-the-art methods already shown are themselves stronger than the older baselines, and DAL consistently outperforms all of them.
>
> **6. Theorem 4.4 / 4.8 discussions.**
>
> In our response to Weakness 8, we described how **we have now moved and expanded these discussions into the main text**.
>
> Specifically, the separation assumption is stated explicitly as its own Assumption 4.6, accompanied by the new Definition 4.5, and followed by Remark 4.7 and an explanatory paragraph. In addition, we have added a new paragraph after Assumption 4.6, that expands on the specifics of the change-point separation. On the other hand, the assumption/requirement on the base algorithm’s regret is also discussed both before, in the paragraph after Remark 4.7, and within the theorem, and mentioned around the same point of the previous Line 414.
>
> We believe this addresses the concern, but we may still be misunderstanding the specific point that feels unclear. If the reviewer could indicate which part of the regret discussion remains confusing (e.g., a particular sentence or transition), we would be very grateful and would gladly revise further.
>
> ---
>
> **We hope that these clarifications help address the reviewer’s remaining concerns. We greatly appreciate the reviewer’s careful reading and ongoing engagement, and we hope that our revisions and explanations may encourage a positive reconsideration of the current score.**

---

### Author Response · Authors · 2025-11-17
**General Response: Revisions and Feedback**

We sincerely thank all reviewers for their thoughtful and constructive feedback, which has greatly improved the quality of our work. Below we summarize the revisions to our draft. The updated version (with changes in blue) and the new experimental code are provided in the latest upload and revised supplementary material.

---

- **Focus on PS**: As noted by Reviewer V4aw, we believe it is more appropriate to state our setting as PS rather than general non-stationary. We have therefore revised the title, abstract, and main text to explicitly reflect that our theoretical setting is PS. The broader “non-stationary” terminology is now used only for motivation and high-level descriptions.

- **Detector Theory and Practice**: Reviewers xZGQ and TeBX requested more details on the non-stationarity detector. In the original submission, these were deferred to the appendix due to space constraints. We would like to highlight that the reason behind DAL's performance is not a specific detector, but rather the theoretical property that the GLR test has. DAL can use any change detector satisfying Property 4.1, not just GLR. Unlike existing detection–restart methods (including MASTER) that hard-wire a detector, DAL is black-box in both the base algorithm and the detector choice. With the additional page, we now (i) clearly state the key detector property in the (new) Property 4.1, (ii) add algorithmic implementations of the GLR and GSR tests as (new) Algorithms 2 and 3, and (iii) move the main theoretical requirements and practical details for the detector into Section 4.1. Finally, we provide the general mathematical formulations of both tests in the appendix.

- **Practical Tuning and Covering Set**: Reviewers fy93 and TeBX raised concerns about the practical tuning of DAL and the construction of the covering set. In our original submission (Section 3), again, due to limited space we had to resort to deferring the details to the Appendix. In the revised version, we add a dedicated Subsection 3.2 that explains how to tune DAL in practice (detectors, base bandit algorithms, and covering sets) in accordance with our theory, while retaining further details in the appendix. In addition, following Reviewer xZGQ’s suggestion, we have revised the proposition for NS-KBs (now Proposition 4.3) to improve clarity.

- **New Experiments**: To demonstrate that DAL’s performance stems from the newly stated Property 4.1 rather than the GLR test itself, we reran our experiments using the GSR detector on both synthetic and real-world benchmarks. The GSR-based DAL performs overall slightly better, supporting our claim that the framework’s success is driven by the underlying detector property. We also improved the contextual-bandit experiments: in the original version, we changed the policy set $\\Pi$ in each run, which led to high variance. In the revised experiments we fix $\\Pi$ across runs, yielding more stable and fair comparisons with the other settings.

- **Clarity of Assumptions**: Reviewers xZGQ and V4aw requested clearer presentation of the assumptions underlying our main theorem. Previously, these were all embedded inside the theorem due to space limitations. We now introduce the separation assumption as a standalone assumption in the new Assumption 4.6, accompanied by Definition 4.5 and followed immediately by both existing and new explanatory text to make its role and implications explicit. We also add further discussion of the assumption on the regret of the input algorithm, including an explanatory paragraph preceding our main Theorem (now Theorem 4.8) and additional clarifications inside the theorem statement.

- **Rigorous Analysis on CBs**: Reviewer TeBX raised concerns about the rigor of our analysis for contextual bandits. In response, we have added a complete, rigorous analysis and proof of our problem formulation (which is based on the contextual-bandit setting) in the appendix. While space constraints prevent us from including a full proof sketch in the main text, we believe this revision strikes a reasonable balance between readability and rigor, and we kindly  reviewers to refer to the revised proof, and we would be happy to further expand this discussion if space permits.

- **Remaining Points**: All other comments and questions from the reviewers have been addressed in detail in the individual responses and incorporated into the revised manuscript to the best of our ability.

---

We sincerely hope to have addressed all of the reviewers' comments to the best of our ability. We would be very happy to receive further feedback if there are additional concerns and hope the reviewers will kindly consider our clarifications and the contributions of our work.

---

### Author Response · Authors · 2025-11-29
**Summary for the new Area Chair**

Given the recent change in the review process and the assignment of a new area chair, we provide a brief, reviewer-anchored summary of our revisions and the discussion with the reviewers.

---

**Scope and title (Reviewers V4aw, xZGQ).**


We clarified that our theoretical results are for piecewise-stationary (PS) bandits, specified the setting to PS accordingly wherever applicable, and also changed the title accordingly. “Non-stationary” is now used only for motivation and for the empirical drifting experiments.

---

**Abstract and base algorithm assumptions (Reviewer xZGQ).**


We refined the abstract and Section 4.2 (above Theorem 4.8) to state precisely that DAL accepts any stationary bandit algorithm with order-optimal $\\tilde{\\mathcal{O}}(\\sqrt{T})$ regret as input. Theorem 4.8 formalizes this requirement and shows how DAL lifts such stationary guarantees to optimal PS regret.

---

**Kernelized Bandit Proposition Revised (Reviwer xZGQ).**


To showcase the connection between the cardinality of $V_T$ and $BL_u$, we revised Proposition 4.3, linking them through $\\delta_T$ in the corresponding bound.

---

**Detector theory and implementation (Reviewers xZGQ, TeBX, fy93).**


We brought the detector details into the main text: a new Property 4.1 specifies the generic condition on any detector; Section 4.1 and Algorithms 2–3 now give the GLR and GSR tests, thresholds, and their role in our theory. Section 2 now briefly explains Line 6 of Alg. 1 and points to these details. We also added experiments with GSR, showing that DAL’s performance is driven by Property 4.1 rather than a specific test and give the implementation details in Section 3.2. Finally, we provide the general mathematical formulation of the GLR and GSR in the Appendix.

---

 **Practical tuning, covering set, and sensitivity (Reviewers fy93, TeBX).**


We added Subsection 3.2 describing in detail how DAL is tuned in practice, i.e., the stationary bandit, the detector parameters, the exploration frequency, and $\mathcal{A}_e$ (covering set), using Propositions 4.2–4.3, Corollary 4.9, and Remark 4.4. We also added a short discussion (Section 3.2) specifically addressing the question *“How sensitive is DAL to the choice of covering set $\mathcal{A}_e$ in large continuous action spaces?”*. We thoroughly explain how our theory translates in practice to assist experimental design. However, due to limited space, we include an extended discussion in the Appendix.

---

**Assumptions regarding the main theorem (Reviewers xZGQ, V4aw).**


We pulled out the change-point separation into a standalone *Assumption 4.6*, accompanied by the new Definition 4.5, a new explanatory paragraph that explains the assumption in more depth and keep the old accompanying explanatory paragraph and Remark immediately after, making its role and implications explicit.

Finally, we clarified the requirement on the base algorithm’s regret before and within Theorem 4.8. A new paragraph in Section 4.2 (after Remark 4.7) connects the informal regret discussion (around the original Line 414) to the formal theorem and the PS regret bounds DAL matches or improves upon.

---

**Contextual Bandit analysis (Reviewer TeBX).**


We updated to a **rigorous CB analysis and full proof** of our problem formulation in the appendix, based on the CB setting underlying our framework.

---

**Experiments and baselines (Reviewer xZGQ).**


We expanded our experimental validation in the synthetic experiments to **include all the additional related algorithms** that were previously only discussed, adding **11 additional algorithms**. To keep the figures interpretable, we group the same paradigms (discounted, budget-restart, and sliding-window families) while giving separate entries to approaches that are meaningfully different. For the real-world experiments, we focus on the current state-of-the-art methods, since the additional algorithms are not competitive with these stronger baselines.

We corrected a high-variance CB setup by fixing the policy class $\Pi$ across runs, and added GLR vs GSR comparisons. On both PS and drifting benchmarks, DAL consistently outperforms all state-of-the-art prior-free methods (MASTER, ADA-OPKB, ADA-ILTCB+) and all prior-based baselines, even with GSR, showcasing DAL’s performance is driven by design rather than a specific detector.

---

To this end, **we have uploaded the final revised version of our draft, with changes marked in blue.** We hope this summary helps the new area chair quickly assess how the revised paper addresses the reviewers’ concerns and clarifies both the theoretical and practical contributions of DAL. For further details, we kindly refer to the revised draft and to our point-by-point responses. We greatly appreciate the AC’s effort in reviewing our work.

---

### Meta-Review · Area_Chair_MpZa · 2025-12-30

**Summary:**

This paper presents a detection-based bandit algorithm to handle piece-wise stationary environments. The claimed benefit is the ability to combine any commonly used bandit algorithm as a black box to handle the non-stationarity. Theoretical analysis is provided against piece-wise stationary environments and extensive experimentations are provided to demonstrate the effectiveness of the proposed solution.

First and foremost, the key issue of this submission is its clear over claim in the first-round submission. As still clearly recorded in the system, the original claim was the solution works with “all common bandit variants” against non-stationarity. But after the rebuttal, the claim was rectified to “any stationary bandit algorithm with order-optimal regret” and “piecewise stationary”. In the rebuttal, the authors defended this change, as “we changed only one word, from “Non-Stationary” to “Piecewise-Stationary,” to more precisely reflect the theoretical scope.” However, this one single word change has completely reshaped the scope of this work: from ANYTHING to a thing.

Second, using an external change detector to handle piecewise stationary environment is not a new idea and it has been well studied in literature. And the assumption imposed in the analysis, i.e., one has enough time to detect the change, has eased most of challenges in the regret analysis, especially when the bandit algorithm itself has order-optimal regret. As a result, the technical contribution of this work is unclear.

Last but not least, there are too many uncommonly used abbreviations in the manuscript, such as PB, NPB, CB, LBs, GLBs. They added unnecessary burden for the audiences to understand the context.

To summarize, the shrinkage of the scope and claims during the rebuttal is unsatisfactory; and even if we only evaluate the rectified scope, its technical contribution against those prior change detection based solution is unclear. Hence, we do not recommend accepting this work to the conference.

**Reviewer Concerns:**

As explained in the meta review, the rebuttal dramatically changed the scope and claim, which has already been questioned by one of the reviewers. Hence, it would be hard to believe the all the reviewers will be satisfied with the arguments in the rebuttal.

**Reviewer Scores:**

It would not be likely.

---

### Decision · Program_Chairs · 2026-01-26

Reject